# Functional role for Cas cytoplasmic adaptor proteins during cortical axon pathfinding

Jason A. Estep[1☉], Alyssa M. Treptow[1☉], Payton A. Rao[2], Patrick Williamson[3], Wenny Wong[2], Martin M. Riccomagno[1,2,3*]

1 Cell, Molecular and Developmental Biology Graduate Program, Department of Molecular, Cell & Systems Biology, University of California, Riverside, California, United States of America, 2 Neuroscience Graduate Program, Department of Molecular, Cell & Systems Biology, University of California, Riverside, California, United States of America, 3 Neuroscience Undergraduate Program, Department of Molecular, Cell & Systems Biology, University of California, Riverside, California, United States of America

☉ These authors contributed equally to this work.
* martinmr@ucr.edu

## Abstract

Proper neural circuit organization requires individual neurons to project to their targets with high specificity. While several guidance molecules have been shown to mediate axonal fasciculation and pathfinding, less is understood about how neurons intracellularly interpret and integrate these cues. Here we provide genetic evidence that the Crk-Associated Substrate (Cas) family of intracellular adaptor proteins is required for proper fasciculation and guidance of two cortical white matter tracts: the Anterior Commissure (AC) and thalamocortical axons (TCAs). Using a *Cas* Triple Conditional Knock Out (*Cas TcKO*) mouse model, we show that Cas proteins are required for proper TCA projection by a non-neuronal cortical cell population. We also demonstrate a requirement of the β1-integrin receptor for TCA projection, similarly in a population of non-neuronal cortical cells. Additional analysis of *Cas TcKO* mutants reveals a role for Cas proteins in AC fasciculation, here within the neurons themselves. This AC fasciculation requirement is not phenocopied in β1-integrin deficient mutants, suggesting that Cas proteins might signal downstream of a different receptor during this axon pathfinding event. These findings implicate Cas proteins as key mediators of cortical axon tract fasciculation and guidance.

## Author summary

In the developing nervous system, neurons extend axons—long projections that relay information to their targets—to establish neural circuits. Axons follow specific pathways directed by extracellular guidance cues, much like street signs direct traffic. While these guidance cues are well studied, how neurons internally interpret and respond to these signals remains unclear. Here, we examine the role of the Crk-Associated Substrate (Cas) family of intracellular adaptor proteins

**Data availability statement:** All relevant data are within the manuscript and its Supporting information files.

**Funding:** HHS | NIH | National Institute of Neurological Disorders and Stroke (NINDS)(https://www.ninds.nih.gov):(M-MR) R01NS139914; California Institute for Regenerative Medicine (CIRM)(https://www.cirm.ca.gov):(AMT) TRANSCEND. The content is solely the responsibility of the authors and does not necessarily represent the official views of the funding agencies. The funders did not play any role in the study design, data collection and analysis, decision to publish, or preparation of the manuscript.

**Competing interests:** The authors have declared that no competing interests exist.

in axon guidance within cortical axon tracts. Using genetic techniques to selectively remove *Cas* gene function from specific cell types, we demonstrate that Cas proteins are required for proper fasciculation (bundling) of anterior commissure axons, acting directly within the projecting axons themselves. Additionally, Cas proteins are required for proper guidance of thalamocortical projections—axons connecting the thalamus with the cortex. However, in this case, Cas proteins do not act within projecting axons but instead direct target neurons to their final positions. We further show that the β1-integrin receptor is similarly required for thalamocortical axon projection. These findings provide genetic evidence for a critical role of Cas adaptor proteins in both fasciculation and guidance of cortical axon tracts.

## Introduction

During Central Nervous System (CNS) development, axons often traverse complex routes to meet their synaptic targets, interacting with a diverse array of attractive, repulsive, and permissive cues [1,2]. One developmental mechanism proposed to ensure proper guidance of large tracts is the concept of "pioneer" versus "follower" axons [3–7]. In this framework, initial pioneering axons carve out stereotyped trajectories due to the presence of intermediate targets, or guidepost cells [8]. These pioneer axons then serve as tracks that subsequent follower axons may use to reach a common target. Both the guidepost-to-guidepost advancement of pioneering axons, as well as the regulated fasciculation and defasciculation of follower axons, require neurons to properly sense and respond to adhesive signals [9,10] in the forms of cell-cell and cell-extracellular matrix (ECM) interactions. In the vertebrate nervous system, two well-established examples of guidepost-assisted axonal guidance are thalamocortical axon (TCA) projections [11,12] and the commissural tracts of the forebrain, such as the Anterior Commissure (AC) [13]. However, the exact cellular and molecular mechanisms that permit the timely interpretation of adhesive signals during TCA guidance or AC formation remain to be elucidated.

During normal development, TCA projections are guided by a diverse array of secreted gradients, guidepost populations, and intermediate targets as they navigate their way from the thalamus to the developing neocortex [14–16]. In mice, TCAs, originating from the thalamus at embryonic day 12 (E12), first turn dorsolaterally at the diencephalic–telencephalic boundary (DTB), then fasciculate to form the internal capsule on E13 and advance through the subpallium until they reach the pallial-subpallial boundary at E14 [12,17]. At the pallial-subpallial boundary, TCAs encounter corticofugal axons and arrest for two to three days before entering the cortical plate, temporarily targeting a cortical region known as the subplate [18–20]. Entry of thalamic axons into the cortical layers and coordinated target selection is facilitated by the subplate, a developmentally transient population of the earliest-born telencephalic neurons that serves as an intermediate target for TCA pathfinding [18,19,21–23]. By E18, TCAs innervate the cortical plate through the intermediate zone and

defasciculate to reach their individual targets [12,15], arriving at the neocortex before their target layer (layer IV) is fully formed. Recent studies suggest that TCAs help promote the specification of layer IV neurons as the cortex develops [24].

The formation of commissural tracts in the forebrain is also known to involve the assistance of additional intermediate targets, often in the form of transient midline glial populations [25–28], such as the glial "tunnel" that facilitates the formation of the Anterior Commissure [29,30]. The AC is composed of two tracts that share a midline crossing point. Beginning at E14-E15 in mice, axons of the anterior AC (aAC) originate from the Anterior Olfactory Nucleus and form reciprocal connections to each olfactory bulb, while axons of the posterior AC (pAC) originate from the ventrolateral cortex and join with the stria terminalis to form reciprocal connections between the two hemispheres [31,32]. Recent developmental studies have identified a population of pioneering axons that precede primary fascicle formation and midline crossing during pAC establishment and suggest a role for ECM molecules in the sorting of aAC and pAC axons [33].

The Crk-Associated Substrate (Cas) family of signal adaptor proteins is comprised of four paralogues: p130Cas (Bcar1), Cas-L (HEF1, NEDD9), Sin (EFS), and Cass4 [34,35]. All four Cas paralogues have high sequence conservation and structure similarity with an N-terminal SH3-domain, a central substrate domain containing 15 repeats of the Tyr-X-X-Pro (YxxP) motif, and a serine-rich domain [35]. Upon upstream receptor activation, Cas proteins are phosphorylated on tyrosine residues within the substrate domain, thus permitting docking of SH2- and PTB-containing partners such as Crk, CrkL, Nck, and SHIPTP2 [34,36–39]. Biochemical studies have demonstrated that Cas proteins are phosphorylated downstream of Integrin [40–42], Dystroglycan [43], CXCL12/CXCR4 [44], Neuropilin [45], and Netrin [46] signaling. Phosphorylation of Cas induces the assembly of the Cas-Crk-DOCK180 scaffold complex which subsequently recruits and activates Rac GTPases, leading to local actin polymerization [34,47,48]. In this way, Cas proteins act as an essential link between external adhesion signals and the actin cytoskeleton, permitting cells to reorganize their actin network in response to ECM engagement.

While p130Cas [36], CasL [49], and Sin [50] are all expressed in the embryonic nervous system, Cass4 expression appears to be restricted primarily to lung, spleen, and blood cells, with minimal brain expression [51]. Within the context of neurodevelopment, Cas proteins have been shown to regulate motor axon fasciculation and projection in Drosophila [41], with similar roles in the fasciculation and pathfinding of Dorsal Root Ganglion (DRG) axons in the mammalian Peripheral Nervous System (PNS) [52]. In the mammalian CNS, Cas proteins play additional roles during neuronal migration and axonal guidance events: Cas adaptor proteins coordinate commissural projections in the spinal cord [46], organize Retinal Ganglion Cell positioning in the mouse retina [42], and have been recently shown to participate in cortical lamination [43].

In this study, we sought to investigate the potential roles of Cas proteins during the formation of forebrain white matter tracts. Accordingly, we focused our investigation on the three Cas family members with established expression and functional roles in neural development (p130Cas, CasL, and Sin). Using conditional reverse genetics, we have found that Cas proteins are required for proper guidance of TCA projections and fasciculation of pAC axons. We demonstrate that the TCA projection phenotype is cortical-autonomous but non-neuronal-autonomous (i.e., Cas genes function in a population of cortical non-neuronal cells to regulate proper lamination), with mis-projecting TCA processes closely associating with misplaced subplate and deep layer neurons. Further, we demonstrate a cortical- and neuronal-autonomous role for Cas genes (i.e., Cas genes function in cortical neurons) during AC fasciculation and show that defasciculating AC axons originate, in part, from the dorsolateral cortex. Interestingly, defects in AC fasciculation are not phenocopied by β1-integrin mutants, suggesting that Cas proteins act, at least in part, independently of β1-integrin during AC formation.

## Results

To investigate the potential roles of Cas family genes during cortical white matter tract formation, we began by assessing the expression of the Cas family paralogues during cortical development. We recently established that transcripts for the paralogues p130Cas, CasL/Nedd9, and Sin/Efs are all expressed in the developing neocortex, with overlapping

expression in the subventricular and ventricular zones [43]. The *p130Cas* transcript is dynamically enriched in the cortical plate at E12.5 and the intermediate zone at E14.5, consistent with expression in postmitotic neurons that are projecting axons [43].

We next performed immunohistochemistry in Swiss Webster embryos at E14.5, E16.5, and E18.5 for p130Cas protein and the cortical white matter marker Neural Cell Adhesion Molecule L1 (NCAM-L1) (Fig 1) [53–55] and the radial glia marker Nestin (S1 Fig) [56,57]. At E14.5, p130Cas signal was detected in the intermediate zone, cortical plate, and marginal zone (Fig 1A–1C'). p130Cas signal co-expressed with the radial glial cell (RGC) marker Nestin in the cortical plate and at the pial surface where radial glial endfeet adhere to the basement membrane (S1A–S1B' Fig). High-magnification images show co-expression of p130Cas with the axonal marker NCAM-L1 in the presumptive Anterior Commissure (AC) (Fig 1B–1B', white arrowhead). At E16.5, p130Cas protein was broadly expressed throughout the developing neocortex, with the most robust signal observed in the marginal zone, subventricular zone, and intermediate zone (Fig 1D–1F'). Additional high-magnification images demonstrate p130Cas signal strongly co-expressed with NCAM-L1 in the External Capsule (EC) (Fig 1E and 1E', white arrow) and the now midline-crossing AC (Fig 1E and 1E', white arrowheads). By E18.5, p130Cas expression was diffusely expressed throughout the developing cortex (Fig 1G–1I'), with stronger expression in the subventricular zone (Fig 1I) and cortical white matter tracts, including the corpus callosum, internal capsule, EC, and AC (Fig 1G). Clear co-expression of p130Cas with NCAM-L1 was observed in the EC (Fig 1H and 1H', white arrow) and AC (Fig 1H, white arrowheads). p130Cas expression continued to overlap with that of Nestin at E16.5 and E18.5 (S1C–S1F' Fig).

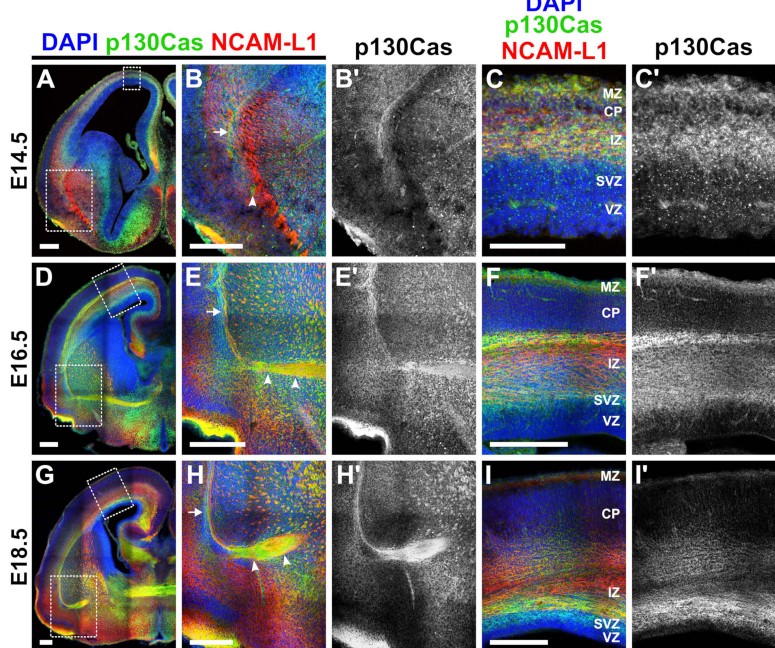

**Fig 1. p130Cas protein is expressed in the developing cortex during lamination and white matter tract formation.** (A-I') Immunohistochemistry showing the developmental expression pattern of p130Cas protein (green) during cortical development of wild-type mice. At E14.5, p130Cas expression is strongly detected in the intermediate zone and marginal zone and more diffusely in the cortical plate (C-C'). At E16.5, p130Cas expression is strongest in the cortical white matter, intermediate zone, subventricular zone, and marginal zone (F-F'). By E18.5, p130Cas is broadly expressed throughout the cortex, with stronger expression in the subventricular, marginal, and intermediate zones (I-I'). Robust co-expression of p130Cas with axonal marker NCAM-L1 (red) is observed in cortical white matter tracts, including the External Capsule (EC) (B, E, and H, white arrowhead) and the Anterior Commissure (AC) (B, E, and H, white arrows). MZ = Marginal Zone; CP = Cortical Plate; IZ = Intermediate Zone; SVZ = Subventricular Zone; VZ = Ventricular Zone. Scale bar = 250μm (A-B', D-I'). Scale bar = 100μm (C-C'). E14.5: n = 4; E16.5: n = 3; and E18.5: n = 4.

To better visualize axons originating from neurons expressing *p130Cas*, we took advantage of the *p130Cas-EGFP-BAC* transgenic line generated by the GENSAT BAC transgenic project [58]. These mice carry a bacterial artificial chromosome (BAC) containing an enhanced green fluorescent protein (EGFP) expression cassette which is driven by the *p130Cas* promoter along with its regulatory sequences, thereby causing all cells with *p130Cas* promoter activity to accumulate cytosolic EGFP in a pattern that recapitulates endogenous gene expression [43,52].

We collected *p130Cas-EGFP-BAC* embryos at E14.5 and E16.5 and neonates at postnatal day 0 (P0) and performed immunohistochemistry for EGFP and NCAM-L1 to visualize cortical white matter tracts (Fig 2). At E14.5, we found *p130Cas*-driven EGFP signal overlapped with NCAM-L1 in multiple areas: the cortical plate, the lateral cortex, and a group of fasciculating axons forming the presumptive AC (Fig 2A–2A", white arrowheads). By E16.5, robust *p130Cas*-driven EGFP signal was observed broadly in the cortical plate and the marginal zone (Fig 2B–2B'). EGFP signal was also detected in white matter, with clear expression in internal capsule axons, the EC (Fig 2B–2B", white arrow), and the midline-crossing AC (Fig 2B, white arrowheads). In neonates (P0), we observed strong *p130Cas*-driven EGFP in all major cortical tracts including the corpus callosum, striatum, EC, and AC (Fig 2C–2C", white arrow denotes EC and white arrowheads denote AC). Higher-magnification images of *p130Cas-EGFP-BAC* neonates at the pallial-subpallial boundary revealed many diffuse NCAM-L1+ processes innervating the subpallium that were not expressing *p130Cas*-driven EGFP, although many EGFP+ cell bodies occupied this region (Fig 2D–2D", white dotted line indicates pallial-subpallial boundary). In the cingulate cortex, high-magnification images of P0 *p130Cas*-EGFP-BAC animals showed robust expression of EGFP in the cortical plate and the corpus callosum (Fig 2E–2E").

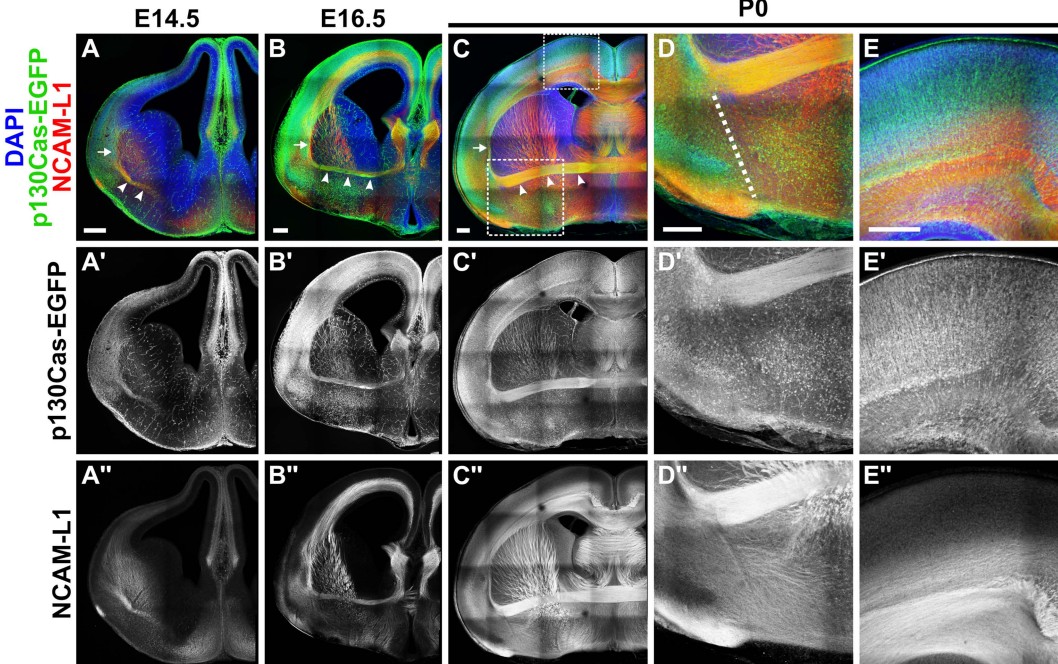

**Fig 2.  *p130Cas* is expressed in neurons that project axons along the cortical white matter.** (A-E") Detection of EGFP (green) expression of *p130Cas-EGFP-BAC* transgenic mice co-immunolabeled with the axonal marker NCAM-L1 (red) at E14.5 (A-A"), E16.5 (B-B"), and P0 (C-E"). Analysis reveals strong *p130Cas* promoter activity during the formation of cortical white matter tracts. Robust EGFP expression is observed in the External Capsule (A, B and C, white arrow) and Anterior Commissure (A, B and C, white arrowheads). High-resolution images at P0 show EGFP (D', E') and NCAM-L1 (D", E") expression at the pallial-subpallial boundary (PSPB) (D, dotted line) and the cortical plate (E). Scale bar = 250µm. n = 3 animals for each developmental timepoint.

Having confirmed the developmental expression of *Cas* genes during the formation of cortical white matter, we next sought to test the requirement of Cas proteins during cortical tract formation. Thus, we generated *Cas* triple conditional knockouts (*Cas TcKO*) [42,43,52] to assess the broad requirements for Cas proteins (p130Cas, CasL, and Sin) during cortical tract development. In brief, *Cas TcKO* animals bear homozygous null alleles for the Cas homologues *CasL* (*CasL*-/-) and *Sin* (*Sin*-/-) while also harboring one floxed allele and one deletion (delta) allele for *p130Cas* (*p130Cas*fl/Δ). Constitutive deletion of *p130Cas* results in early embryonic lethality due to defects in cardiovascular development [59]. Thus, crossing TcKO animals to cell-specific Cre-driver lines permits Cre-mediated excision of *p130Cas* in neural tissue only, allowing embryos to survive through development.

To determine the requirement of *Cas* genes during cortical white matter tract formation, we chose to first induce a broad deletion of *Cas* gene function by crossing *Cas TcKO* animals with *Nestin-Cre* mice. *Nestin-Cre* drives early (E9.5) recombination in neural stem cells and intermediate progenitors [60]. As such, *Nestin-Cre* is often employed to produce CNS-wide deletions in both neuronal and glial populations. We generated *Nestin-Cre;CasL*-/-*;Sin*-/-*;p130Cas*fl/Δ (or more simply *Nes-Cre;TcKO*) mutants and started by confirming the specificity of the p130Cas antibody by immunostaining control (*CasL*-/-*;Sin*-/-*;p130Cas*fl/+) and littermate *Nes-Cre;TcKO* brains at E16.5 (S2 Fig). The p130Cas antibody showed strong specificity for the EC-AC tract and the cortical plate, with staining only remaining in *Nes-Cre;TcKO* animals in a small subpopulation of cells in the hypothalamus (S2 Fig). We next performed immunohistochemistry at P0 for the axon marker NCAM-L1 to examine the organization of major cortical tracts (Fig 3). We observed notable disruptions to two major white matter tracts in *Nes-Cre;TcKO* mutants: the Anterior Commissure (AC) and cortical white matter.

Axons of the External Capsule (EC) that, upon exit from the pallium, normally turn medially to join the posterior Anterior Commissure (pAC), fail to fasciculate properly in *Nes-Cre;TcKO* mutants, and instead project ventrally into the subpallium (Fig 3A, 3A', 3B and 3B'). Tracing experiments using DiA crystals placed into the Somatosensory 2 (S2) region of the cortex (S3A–S3C Fig) confirmed that these mis-projecting axons originate in the dorsolateral and lateral cortex (Fig 3C and 3D; n = 6 for *Nes-Cre;TcKO,* n = 7 for controls). We quantified axon pathfinding errors by calculating the average number of stray pAC axon bundles per section per animal in the subpallial area in *Nes-Cre;TcKO* mutants and control mice. The number of stray axon fascicles in the subpallial area in *Nes-Cre;TcKO* animals averaged 4.23 ± 1.12, while the average was 0 for control mice (Fig 3I; S1 Table; Unpaired *t*-test, p = 0.0004, n = 7 for *Nes-Cre;TcKO,* n = 10 for controls).

*Nes-Cre;TcKO* mutants also display severe defects in white matter within the cortical plate, with large bundles of axons mis-projecting towards the pial surface (Fig 3A'' and 3B''). We hereafter refer to these ectopic projections as Cortical Bundles (CBs). To quantify the CB phenotype, we calculated the average number of CBs present per section per animal in the cortex of both *Nes-Cre;TcKO* mutants and control mice. On average, we observed 7.88 ± 0.72 CBs in the cortex of *Nes-Cre;TcKO* and 0 in littermate controls (Fig 3J; S1 Table; Unpaired *t*-test, p < 0.0001, n = 8 for *Nes-Cre;TcKO,* n = 7 for controls). The cortex receives input from multiple cortical and subcortical areas, including contralateral connections via the corpus callosum and AC [8] and thalamic connections from thalamocortical axon (TCA) projections [61,62]. We therefore asked if the CB phenotype presented by *Nes-Cre;TcKO* mutants consisted of axons originating from cortical or subcortical areas. Initial tracing experiments using DiA crystals placed into the S2 cortex indicated that mis-projecting axons in CBs are not cortical in origin (Fig 3E–3F). We performed additional lipophilic tracing experiments by placing DiI crystals into the laterodorsal nucleus of the thalamus (S3D–S3F Fig) and found robust labeling of CBs in the ipsilateral hemisphere in *Nes-Cre;TcKO* mutant neonates (Fig 3G and 3H; n = 6 for *Nes-Cre;TcKO,* n = 5 for controls). This suggests that CBs form primarily from TCA afferents. Taken together, these data point to a strong requirement for *Cas* family genes during cortical white matter tract formation.

To examine the developmental progression of *Nes-Cre;TcKO* white matter tract defects, we collected embryos at E14.5, E16.5, and E18.5 and performed immunohistochemistry for NCAM-L1 (Fig 4). Interestingly, we observed different developmental trajectories for each mutant phenotype. Initial formation of the EC-AC tract was unperturbed in *Nes-Cre;TcKO* animals with no observable axon defasciculation at E14.5 (Fig 4A, 4A', 4B and 4B'; n = 4 for *Nes-Cre;TcKO,* n = 4 for

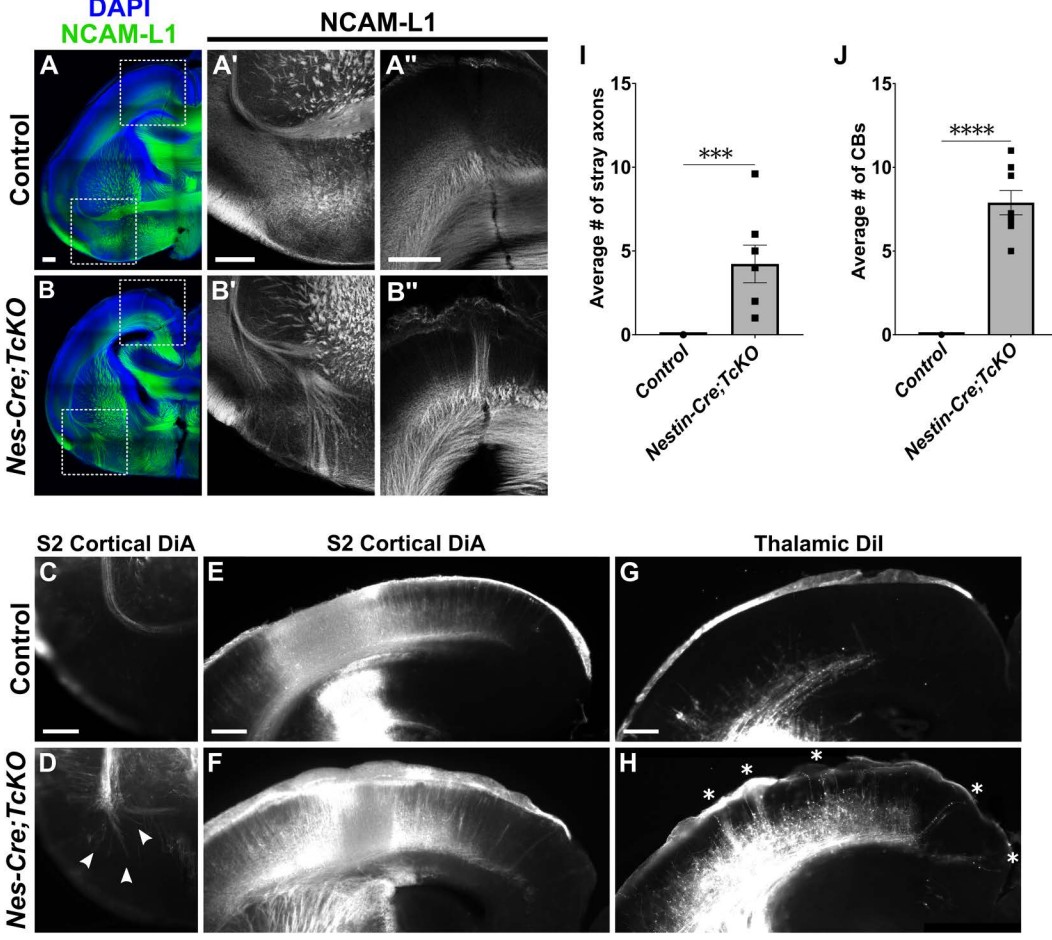

**Fig 3. *Cas TcKO* mutants display errors in cortical and thalamocortical connections.** (A-B") Immunohistological analysis of major cortical tracts using the axonal marker NCAM-L1 (green) in *Nes-Cre;TcKO* mutant and control neonates (P0). Two major defasciculation phenotypes are present in *Nes-Cre;TcKO* mice: 1) efferents originating in the cortex fail to fasciculate properly in the pAC tract and erroneously project into the subpallium (B'), and 2) thalamocortical connections meant to innervate layer IV mis-project to the subpial boundary, forming large Cortical Bundles (B"). Each of these phenotypes is fully penetrant in *Nes-Cre;TcKO* mutants. The high-magnification panel in B" is from a similar area as the upper box in B, but from a nearby section. (C-H) The origin of mis-projecting axons is confirmed via anterograde lipophilic tracing experiments. DiA crystals placed into the S2 cortex label ventrally defasciculating AC axons (D, arrowheads) but not Cortical Bundles (F) in *Nes-Cre;TcKO* mutants. DiI crystals placed in the laterodorsal thalamus label Cortical Bundles (H, white asterisks). (I) Quantification of the number of stray axon bundles in the subpallial area. Unpaired *t*-test, ***p < 0.001, n = 7 for *Nes-Cre;TcKO,* n = 10 for controls. (J) Quantification of the number of Cortical Bundles. Unpaired *t*-test, ****p < 0.0001, n = 8 for *Nes-Cre;TcKO,* n = 7 for controls. Values provided are mean ± SEM. Scale bar = 250μm.

controls) and E16.5 (Fig 4C, 4C', 4D and 4D'; n = 8 for *Nes-Cre;TcKO,* n = 12 for controls). It was not until late embryonic development (E18.5) that we detected aberrant axon defasciculation in the AC (Fig 4E, 4E', 4F and 4F', white arrowhead), with *Nes-Cre;TcKO* mutants averaging 6.55 ± 1.09 stray axon bundles in the subpallial area, while control animals averaged 0 (Fig 4G; S1 Table; Unpaired *t*-test, p = 0.0011, n = 5 for *Nes-Cre;TcKO,* n = 4 for controls). In contrast, the CB phenotype is apparent in *Nes-Cre;TcKO* mutants as early as E14.5 (Fig 4A" and 4B", white asterisks denote CBs), with the phenotype growing in severity as development continued (Fig 4C", 4D", 4E" and 4F", white asterisks denote CBs). The average number of CBs in *Nes-Cre;TcKO* mutant mice increased with age (4.29 ± 0.79 vs 5.06 ± 0.23 vs 9.06 ± 0.76), and at each embryonic time point, the number of CBs observed on average in *Nes-Cre;TcKO* mutants was significantly

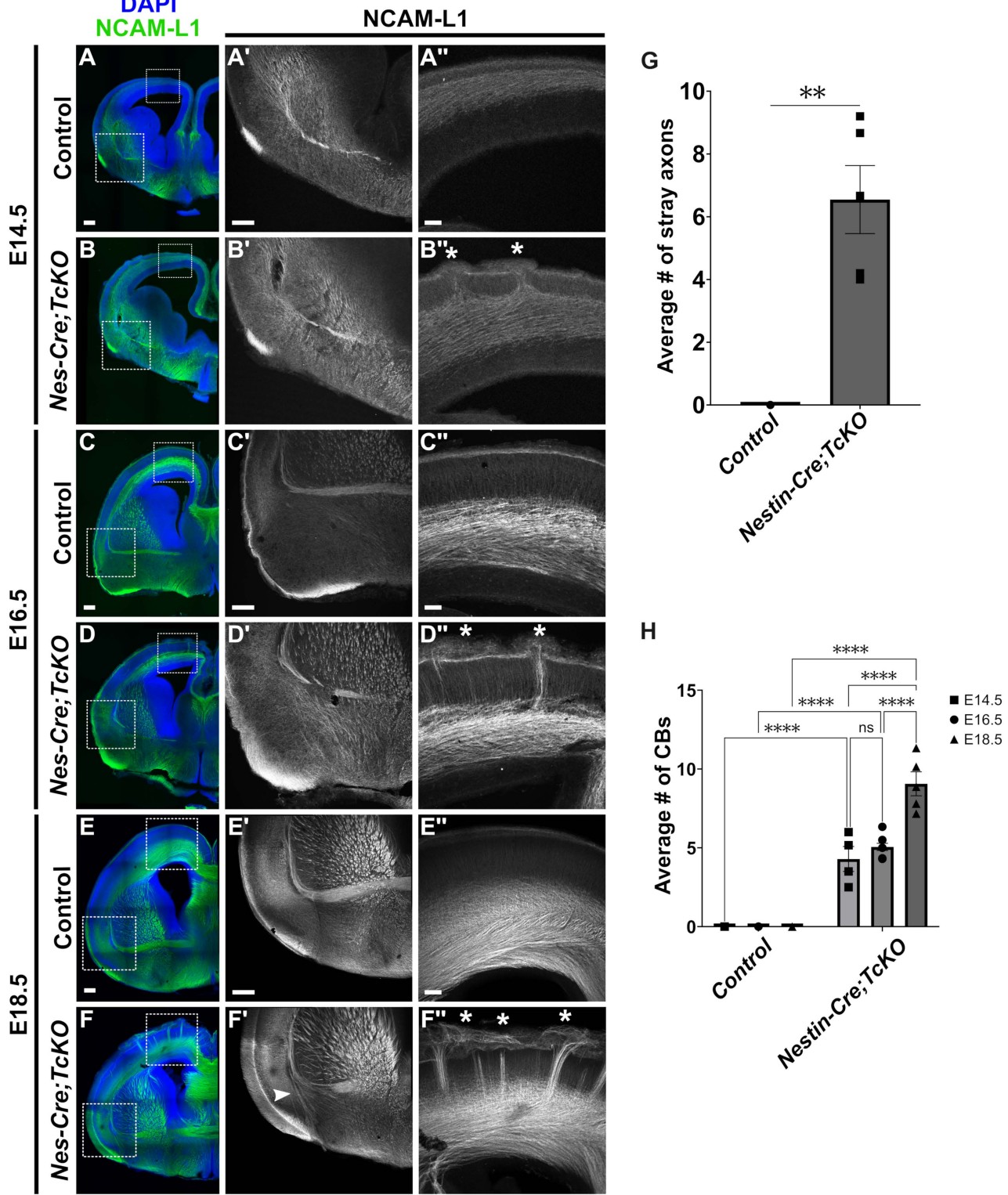

**Fig 4. Developmental progression of white matter defects in *Nes-Cre;TcKO* mutants.** (A-F") Immunohistochemistry for the axonal marker NCAM-L1 (green) in *Nes-Cre;TcKO* mutants and littermate control mice at E14.5 (A-B"), E16.5 (C-D"), and E18.5 (E-F"). Analysis reveals different developmental timelines for each defasciculation phenotype. The EC-AC tract begins developing at E14.5 in control (A') and *Nes-Cre;TcKO* mice (B').

*Nes-Cre;TcKO* mice display no AC defasciculation errors until E18.5 (F', white arrowhead). Cortical Bundles are apparent at E14.5 in *Nes-Cre;TcKO* mutants (B", white asterisks) and worsen in severity as development continues (D" and F", white asterisks). (G) Quantification of the average number of stray axon bundles in the subpallial area at E18.5. Unpaired *t*-test, **p < 0.01, n = 5 for *Nes-Cre;TcKO,* n = 4 for controls. (H) Quantification of the average number of Cortical Bundles at E14.6, E16.5, and E18.5. Two-way ANOVA, p < 0.0001; Tukey HSD post hoc test, ****p < 0.0001 E14.5 *Nes-Cre;TcKO* versus control, E16.5 *Nes-Cre;TcKO* versus control, E18.5 *Nes-Cre;TcKO* versus control, E14.5 *Nes-Cre;TcKO* versus E18.5 *Nes-Cre;TcKO*, E16.5 *Nes-Cre;TcKO* versus E18.5 *Nes-Cre;TcKO*; ns E14.5 *Nes-Cre;TcKO* vs E16.5 *Nes-Cre;TcKO*; E14.5: n = 4 for *Nes-Cre;TcKO,* n = 4 for controls; E16.5: n = 8 for *Nes-Cre;TcKO,* n = 12 for controls; E18.5: n = 5 for *Nes-Cre;TcKO,* n = 4 for controls). Values provided are mean ± SEM. Scale bar = 250μm.

higher compared to littermate controls (Fig 4H; S1 Table; two-way ANOVA, p < 0.0001; Tukey honestly significant difference (HSD) post hoc test, p < 0.0001 E14.5 *Nes-Cre;TcKO* versus control, E16.5 *Nes-Cre;TcKO* versus control, E18.5 *Nes-Cre;TcKO* versus control, E14.5 *Nes-Cre;TcKO* versus E18.5 *Nes-Cre;TcKO*, E16.5 *Nes-Cre;TcKO* versus E18.5 *Nes-Cre;TcKO*; ns E14.5 *Nes-Cre;TcKO* vs E16.5 *Nes-Cre;TcKO*; E14.5: n = 4 for *Nes-Cre;TcKO,* n = 4 for controls; E16.5: n = 8 for *Nes-Cre;TcKO,* n = 12 for controls; E18.5: n = 5 for *Nes-Cre;TcKO,* n = 4 for controls). These results suggest different developmental trajectories for each of the presented white matter tract mutant phenotypes.

We next asked if *Cas* family genes are functioning in a cortical-autonomous and neuronal-autonomous manner during establishment of the EC-AC and TCA tracts. With this aim, we generated two new classes of *Cas TcKO* mutants by crossing TcKO animals into *Emx1-Cre* [63] and *Nex-Cre* [64] driver lines. In contrast to the broad CNS expression of *Nestin-Cre*, *Emx1-Cre* and *Nex-Cre* recombination events are largely restricted to the cortex, allowing us to assess for a possible cortical-autonomous role of *Cas* genes. *Emx1-Cre* induces recombination in cortical radial glial cells (RGCs), affecting RGCs, cortical excitatory neurons, and macroglial populations, while *Nex-Cre* is active only in early postmitotic excitatory neurons within the cortex. By comparing potential phenotypic differences between *Emx1-Cre;TcKO* and *Nex-Cre;TcKO* mutant animals, we can evaluate if *Cas* genes function non-neuronal autonomously (i.e., in cortical non-neuronal cells) or neuronal-autonomously (i.e., in cortical neurons) within the cortex (S4 Fig).

We have previously validated that *Emx1-Cre;TcKO* and *Nex-Cre;TcKO* mutant animals do not produce functional *Cas* transcripts in the expected neural populations [43]. We then generated *Emx1-Cre;CasL^{-/-};Sin^{-/-};p130Cas^{fl/Δ}* (*Emx1-Cre;TcKO*) mutants and collected neonates (P0) for immunohistochemistry using the axon marker NCAM-L1 (Fig 5A–5B"). We observed defasciculation of AC axons in *Emx1-Cre;TcKO* animals, which phenocopied the AC defasciculation errors in *Nes-Cre;TcKO* neonates (Fig 5A' and 5B', white arrowheads). These defasciculating AC axons were labeled by lipophilic tracing from DiA crystals that were placed in the S2 region of the cortex (Fig 5C and 5D; n = 4 for *Emx1-Cre;TcKO,* n = 5 for controls). In *Emx1-Cre;TcKO* mutants, we observed an average of 7.90 ± 1.43 stray axon bundles in the subpallial area, while in control mice we observed 0 (Fig 5M; S1 Table; Unpaired *t*-test, p < 0.0001, n = 8 for *Emx1-Cre;TcKO,* n = 10 for controls). *Emx1-Cre;TcKO* animals also presented the ectopic CB phenotype, with large bundles of axons mis-projecting to the pial surface (Fig 5A" and 5B", white asterisks). On average, we observed 9.15 ± 0.43 CBs in *Emx1-Cre;TcKO* mice and 0 in controls (Fig 5N; S1 Table; Unpaired *t*-test, p < 0.0001, n = 8 for *Emx1-Cre;TcKO,* n = 10 for controls). Lastly, we performed axon tracing experiments by placing DiI crystals into the thalamus of *Emx1;TcKO* neonates and confirmed that these CBs are primarily formed by thalamic axons (Fig 5E and 5F; n = 3 for *Emx1-Cre;TcKO,* n = 3 for controls).

Our initial analysis showed that the CBs in *Nes-Cre;TcKO* mutants are primarily TCA afferents originating from the thalamic nuclei (Fig 3G and 3H). However, *Emx1-Cre* is not expressed in the thalamic territory [63]. Thus, while the observation of CBs in *Emx1-Cre;TcKO* mutants does not fully exclude the possibility of a thalamic role for Cas genes in TCA innervation, it strongly suggests a cortical-autonomous role for Cas in guiding those axons, thereby demonstrating that the white matter tract phenotypes in *Cas TcKO* animals require Cas gene function in the developing neocortex itself.

Next, we generated *Nex-Cre;CasL^{-/-};Sin^{-/-};p130Cas^{fl/Δ}* (*Nex-Cre;TcKO*) mice and collected neonates (P0) for immunohistochemistry using the axon marker NCAM-L1 (Fig 5G–5H"). In contrast to *Emx1-Cre;TcKO* mutants and *Nestin-Cre;TcKO* mutants, *Nex-Cre;TcKO* mutants only presented the AC defasciculation phenotype (Fig 5G' and 5H', white arrowheads), demonstrating a neuronal-autonomous role for *Cas* genes during AC fasciculation. These stray AC axons were again

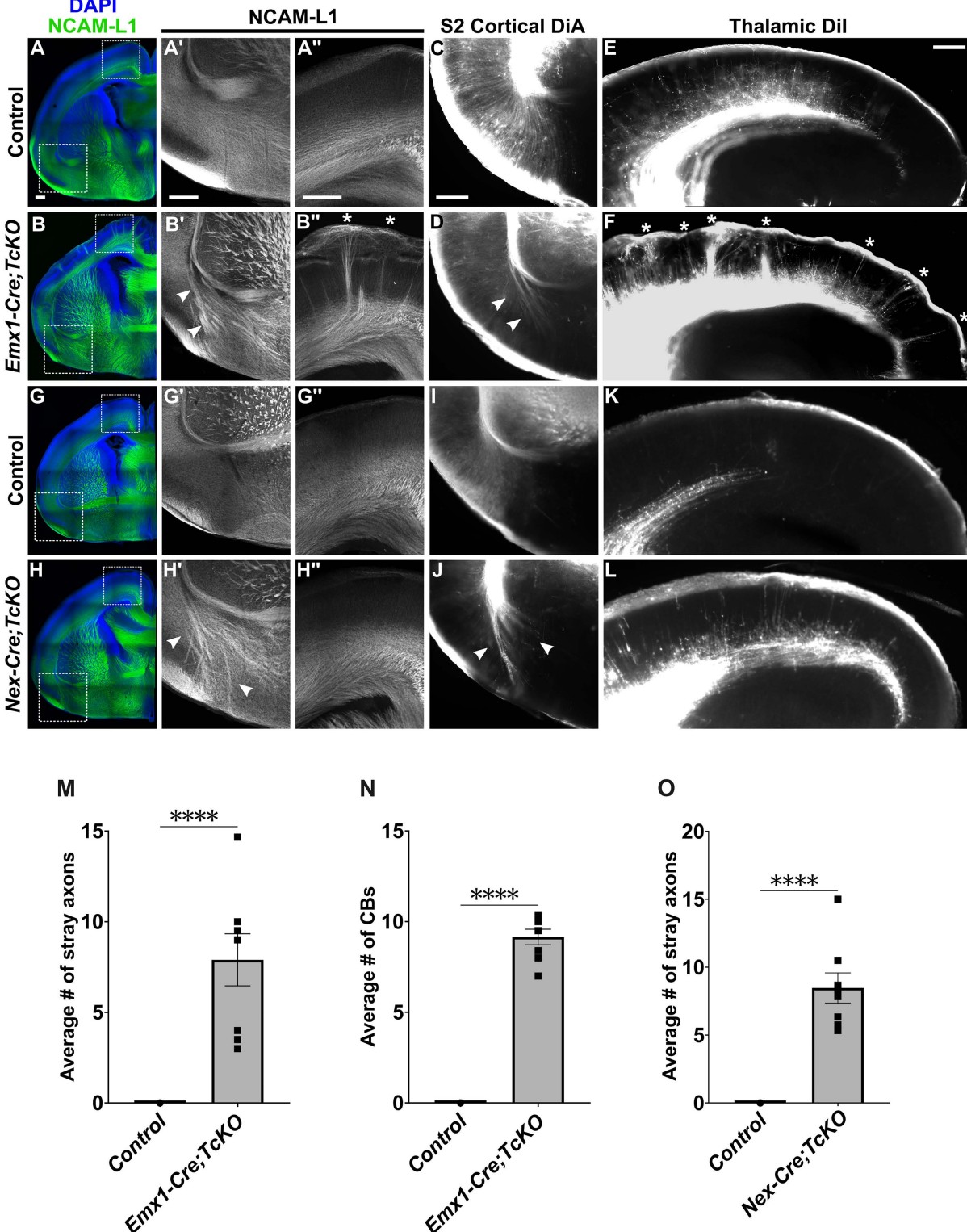

**Fig 5. Neuronal-autonomous and non-neuronal autonomous roles for *Cas* genes in white matter tract fasciculation.** (A-H") Immunohistochemistry for the axonal marker NCAM-L1 (green) in *Emx1-Cre;TcKO* and *Nex-Cre;TcKO* neonates. Analysis reveals both classes of conditional knockouts present the AC defasciculation phenotype (B' and H', white arrowheads), indicating this phenotype is neuronal-autonomous (i.e., Cas proteins function

within the neurons themselves). This phenotype is fully penetrant in *Emx1-Cre;TcKO* and *Nex-Cre;TcKO* mutants. Similar to *Nes-Cre;TcKO* mutants (Fig 3), defasciculating axons in the subpallial area of *Emx1-Cre;TcKO* and *Nex-Cre;TcKO* mutants are cortical in origin as revealed by anterograde DiA tracing from crystal placement in the S2 cortex (D and J, white arrowheads). In contrast, only *Emx1-Cre;TcKO* mutants display the Cortical Bundle phenotype (B", white asterisks). Cortical Bundles are confirmed to be TCA projections by anterograde labeling from DiI crystals placed in the laterodorsal thalamus (F, white asterisks). The absence of the Cortical Bundle phenotype in *Nex-Cre;TcKO* mutants (H" and L) indicates *Cas* genes function in a population of cortical non-neuronal cells during TCA fasciculation and projection. (M) Quantification of the number of stray axon bundles in the subpallial area in *Emx1-Cre;TcKO* mutants and controls. Unpaired *t*-test, ****p < 0.0001, n = 8 for *Emx1-Cre;TcKO*, n = 10 for controls. (N) Quantification of the number of Cortical Bundles in *Emx1-Cre;TcKO* mutants and controls. Unpaired *t*-test, ****p < 0.0001, n = 8 for *Emx1-Cre;TcKO*, n = 10 for controls. (O) Quantification of the number of stray axon bundles in the subpallial area of *Nex-Cre;TcKO* mutants and controls. Unpaired *t*-test, ****p < 0.0001, n = 8 for *Nex-Cre;TcKO*, n = 7 for controls. Values provided are mean ± SEM. Scale bar = 250μm.

confirmed to be dorsolateral and lateral in origin by lipophilic tracing from DiA crystal placement into the S2 cortex (Fig 5I and 5J, white arrowheads; n = 5 for *Nex-Cre;TcKO*, n = 4 for controls). We observed an average of 8.47 ± 1.11 stray axon bundles in the subpallial area of *Nex-Cre;TcKO* mutants and 0 in control mice (Fig 5O; S1 Table; Unpaired *t*-test, p < 0.0001, n = 8 for *Nex-Cre;TcKO*, n = 7 for controls). Interestingly, *Nex-Cre;TcKO* mutant neonates do not present the CB phenotype, indicating a non-neuronal autonomous role of *Cas* genes in TCA projections (Fig 5G", 5H", K and L; n = 8 for *Nex-Cre;TcKO*, n = 7 for controls).

Analyses of the three classes of *Cas TcKO* mutants thus indicate a complex requirement for *Cas* genes during cortical white matter tract formation. The proper fasciculation of AC axons appears to require *Cas* in a neuronal-autonomous manner, likely reflecting a requirement for *Cas* genes in the projecting axons themselves. In contrast, the proper projection of TCA afferents into the cortex appears to have a non-neuronal-autonomous requirement for *Cas* genes. As such, we infer that *Cas* genes are required by some non-neuronal cortical cell population to properly coordinate TCA guidance, possibly by controlling the positioning of intermediate target populations or the provision of local guidance cues in the cortex (S4 Fig).

To further assess the cortical- and neuronal-autonomous role of Cas genes during AC fasciculation, we genetically labeled AC axons to confirm their cortical origins. This facilitated a closer examination of their relationship to neighboring tracts. To this end, we made use of the *Ai14/R26^(LSL-tdTomato)* Cre reporter line (or simply *Ai14*), which expresses tdTomato fluorescent protein upon Cre-mediated excision of upstream translational STOP sequences, permitting visualization of cells or tissues that have or previously had Cre recombinase expression [65]. We crossed the *Ai14* reporter line into the *Emx1-Cre;TcKO* line to generate *Emx1-Cre;Ai14^(+/-);TcKO* mutants and collected P0 neonates for immunohistochemistry with the marker NCAM-L1 (Fig 6A–6D"). As expected, we found robust labeling of the EC-AC tract in *Emx1-Cre;Ai14^(+/-)* controls (*Emx1-Cre;Ai14^(+/-);CasL^(+/-);Sin^(+/-);p130Cas^(+/+)*) and observed tight fasciculation of the AC tract in these animals (Fig 6A–6B). Examination of *Emx1-Cre;Ai14^(+/-);TcKO* mutants revealed that 100% of the large, defasciculating AC axon bundles were tdTomato+, genetically confirming the cortical origin of these mis-projecting axons (Fig 6D–D", white arrowheads). On average, we observed 9.5 ± 0.29 stray axon bundles in the subpallial area of *Emx1-Cre;Ai14^(+/-);TcKO* mice and 0 in *Emx1-Cre;Ai14^(+/-)* controls (Fig 6I; S1 Table; Unpaired *t*-test p < 0.0001, n = 3 for *Emx1-Cre;Ai14^(+/-);TcKO*, n = 3 for controls). Interestingly, we also observed tdTomato-/NCAM-L1+ processes that diffusely populated the subpallium. The thicker bundles of tdTomato+/NCAM-L1+ processes appeared to intercalate with these tdTomato-/NCAM-L1+ processes (Fig 6D), suggesting that Cas-deficient tdTomato+ axons might be improperly associating with other axonal processes.

We next crossed the *Ai14* reporter line into the *Nex-Cre;TcKO* line to generate *NexCre;Ai14^(+/-);TcKO* mutants and collected P0 neonates for immunohistochemistry with the marker NCAM-L1 (Fig 6E–6H"). As observed with *Emx1-Cre;Ai14^(+/-)* controls, we found robust labeling of the EC-AC tract in *Nex-Cre;Ai14^(+/-)* controls (*Nex-Cre;Ai14^(+/-);CasL^(+/-);Sin^(+/-);p130Cas^(+/+)*) (Fig 6E–6F). In *Nex-Cre;Ai14^(+/-);TcKO* mutants, we observed large, defasciculating AC axon bundles in the subpallial area, 100% of which were tdTomato+ (Fig 6H–6H", white arrowheads). On average, we observed 9.33 ± 2.03 stray axon bundles in the subpallial area in *Nex-Cre;Ai14^(+/-);TcKO*, while in *Nex-Cre;Ai14^(+/-)* controls, we observed 0 (Fig 6J; S1 Table; Unpaired *t*-test p = 0.0116, n = 4 for *Nex-Cre;Ai14^(+/-);TcKO*, n = 3 for controls). As in *Emx1-Cre;Ai14^(+/-);TcKO* mutants, the defasciculating tdTomato+/

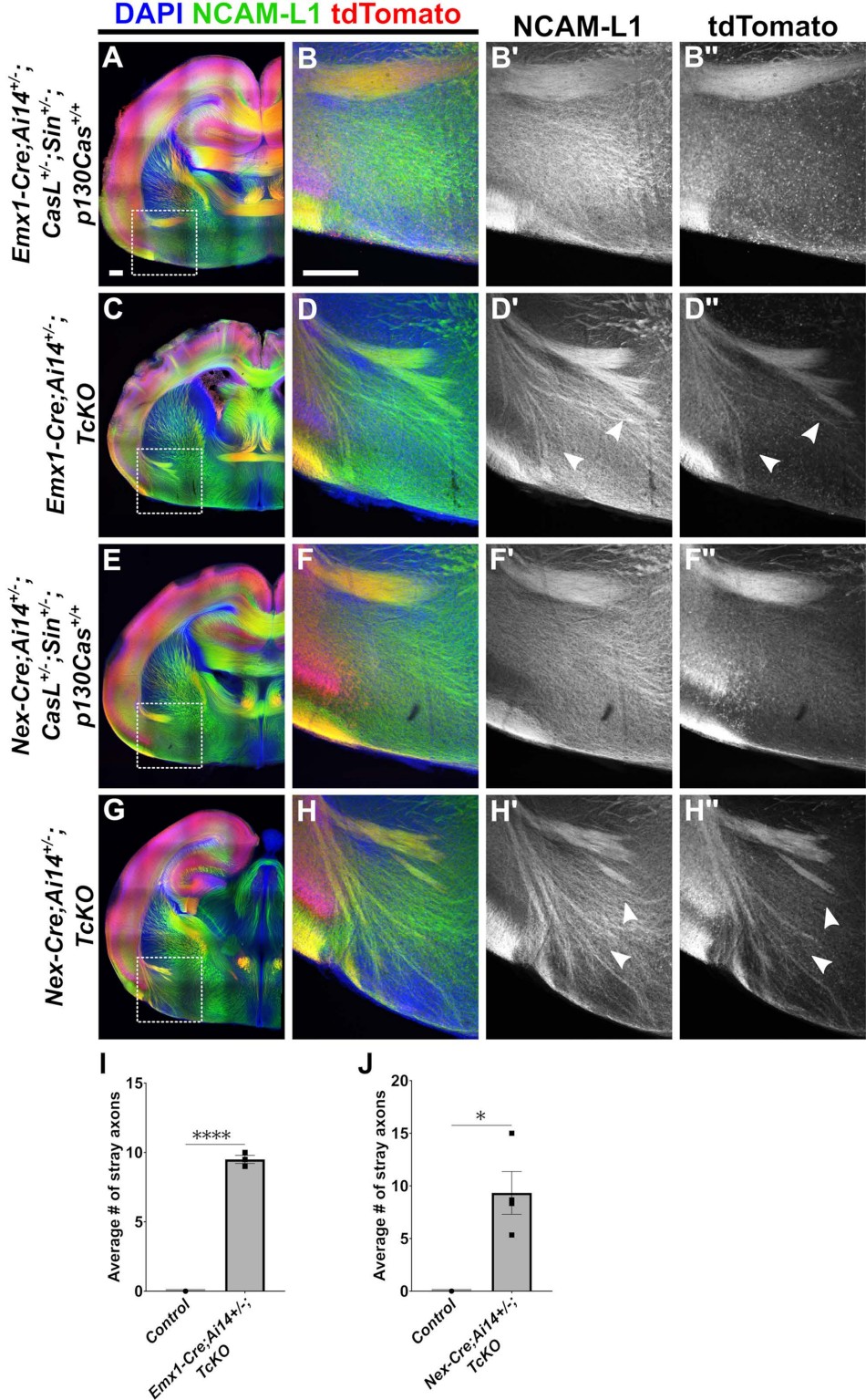

**Fig 6. Genetic labeling confirms cortical origins of AC defasciculating axons in *Cas TcKO* mutants.** (A-H") Immunohistochemistry for NCAM-L1 (green) and the tdTomato fluorescent Cre-dependent reporter (red). Analysis reveals extensive co-labeling in defasciculating AC axon bundles in both *Emx1-Cre;Ai14+/-;TcKO* (C-D") and *Nex-Cre;Ai14+/-;TcKO* (G-H") mutants. In both classes of conditional mutants, tdTomato+ defasciculating axons

appear to intercalate with diffuse NCAM-L1+processes that occupy the subpallium (D' vs D" and H' vs H", white arrowheads). No tdTomato+ processes may be seen invading the subpallium in *Emx1-Cre;Ai14+/-;CasL+/-;Sin+/-;p130Cas+/+* controls (A-B") and *Nex-Cre;Ai14+/-;CasL+/-;Sin+/-;p130Cas+/+* (E-F") controls. (I) Quantification of the number of stray axon bundles in the subpallial area of *Emx1-Cre;Ai14+/-;TcKO* mutants and controls. Unpaired *t*-test, ****p < 0.0001, n = 3 for *Emx1-Cre;Ai14+/-;TcKO*, n = 3 for controls. (J) Quantification of the number of stray axon bundles in the subpallial area of *Nex-Cre;Ai14+/-;TcKO* mutants and controls. Unpaired *t*-test, *p < 0.05, n = 4 for *Nex-Cre;Ai14+/-;TcKO*, n = 3 for controls. Values provided are mean ± SEM. Scale bar = 250μm.

NCAM-L1+AC processes in *Nex-Cre;Ai14+/-;TcKO* mutants appeared to intercalate with the tdTomato-/NCAM-L1+bundles that also populate the subpallium (Fig 6H). Taken together, these data genetically confirm the cortical origins of defasciculating axons in the AC while also supporting the neuronal-autonomous role of *Cas* genes for proper AC fasciculation.

During canonical adhesion signaling events, Cas proteins are recruited to early Integrin Adhesion Complexes (IACs) via β1-integrin receptor stimulation and subsequent activation of the non-receptor tyrosine kinases Focal Adhesion Kinase (FAK) and Src [34]. In the mammalian brain, β1-integrin, encoded by *Itgb1*, appears to be the most prevalent β subunit involved in mediating adhesion signaling events [66–70], with established roles in axonal guidance and growth cone dynamics [67,71–73]. It has been demonstrated that *Itgb1* is expressed in the developing cortex [74,75]. We attempted to validate β1-integrin protein expression in the developing cortex using two different β1-integrin antibodies, but neither demonstrated specificity (S5 and S6 Figs). To investigate whether β1-integrin could be the adhesion receptor acting upstream of Cas proteins during cortical white matter tract development, we asked if the requirement for *Cas* genes in AC fasciculation and TCA projection was also observed in *Itgb1* mutants.

We therefore generated *Emx1-Cre;Itgb1fl/fl* and *Nex-Cre;Itgb1fl/fl* mice and performed immunohistochemistry in neonates (P0) using the axon marker NCAM-L1 (Fig 7). We first examined the AC commissure tract in *Emx1-Cre;Itgb1fl/fl* and *Nex-Cre;Itgb1fl/fl* neonates. Remarkably, neither class of *Itgb1* mutants phenocopied the AC defasciculation defects observed in *Cas TcKO* mutants (Fig 7A, 7A', 7B, 7B', 7C, 7C', 7D and 7D'; n = 6 for *Emx1-Cre;Itgb1fl/fl,* n = 5 for controls; n = 3 for *Nex-Cre;Itgb1fl/fl* mutants, n = 4 for controls). These observations demonstrate that the β1-integrin receptor by itself is dispensable for AC fasciculation, suggesting a different receptor may be acting upstream of *Cas* genes during AC formation. We next examined the cortices of *Emx1-Cre;Itgb1fl/fl* and *Nex-Cre;Itgb1fl/fl* mutants and observed bundles of mis-projecting axons in *Emx1-Cre;Itgb1fl/fl* mutants (Fig 7A" and 7B", white asterisks), a defect that strongly phenocopied the CB phenotype presented by *Emx1-Cre;TcKO* mutants (Fig 5). On average, we observed 8.08 ± 0.25 CBs in *Emx1-Cre;Itgb1fl/fl* animals and 0 in controls (Fig 7E; S1 Table; Unpaired *t*-test, p < 0.0001, n = 6 for *Emx1-Cre;Itgb1fl/fl,* n = 5 for controls). Interestingly, *Nex-Cre;Itgb1fl/fl* mutants did not display Cortical Bundles (Fig 7C" and 7D"; n = 3 for *Nex-Cre;Itgb1fl/fl* mutants, n = 4 for controls). This suggests similar non-neuronal-autonomous roles for β1-integrin and Cas proteins during TCA projection pathfinding (S4 Fig).

Given that β1-integrin seems to be dispensable on its own for AC formation, we sought to determine the expression of other β-integrin subunits previously reported to be expressed during cortical development [75]. Transcripts for the β-integrin subunits β5-, β6-, and β8-integrin (encoded by *Itgb5*, *Itgb6*, and *Itgb8*, respectively) are indeed expressed in the developing neocortex at E14.5 (S7 Fig), E16.5 (S8 Fig), and E18.5 (S9 Fig), with overlapping expression in the ventricular zone at E18.5 (S9 Fig). Moreover, throughout development, *Itgb6* and *Itgb8* are diffusely expressed in the piriform cortex (S7–S9 Figs), where neurons that project axons that contribute to the formation of the pAC reside.

Lastly, we sought to identify a developmental mechanism by which TCA afferents mis-project to form CBs in *Cas TcKO* mutants. Previous research from our lab established a role for Cas proteins and β1-integrin during cortical lamination [43]. We therefore asked if the CB phenotype presented by *Emx1-Cre;TcKO* and *Emx1-Cre;Itgb1fl/fl* mutants was a secondary phenotype that resulted from the disruption of cortical lamination in these mutants. As previously mentioned, the subplate is a transient structure that acts as an intermediate target for TCAs, coordinating the entry of these thalamic axons into the cortical plate [22,61]. Given the cortical requirement of *Cas* genes for TCA guidance, and the defects in cortical lamination presented by *Cas TcKO* mutants [43], we hypothesized that one possible explanation for the CB phenotype presented

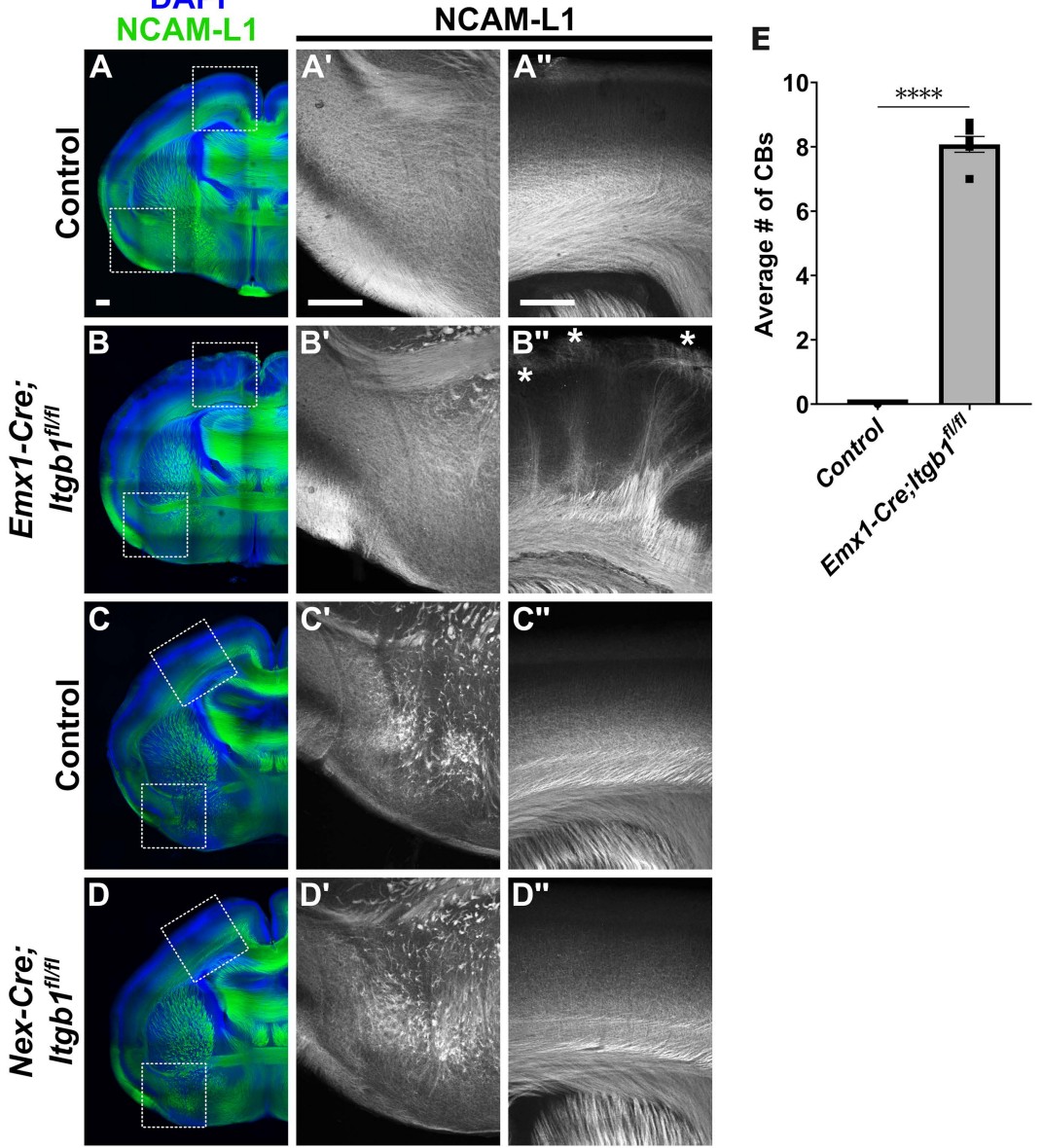

**Fig 7. *Itgb1* mutants phenocopy the TCA tract defects observed in *Cas TcKO* mutants.** (A-D") Immunohistological analysis of NCAM-L1 (green) in *Emx1-Cre;Itgb1^fl/fl^* and *Nex-Cre;Itgb1^fl/fl^* neonates. Neither class of conditional mutant phenocopies the AC defasciculation defects observed in *Cas TcKO* mutants (B' and D'), suggesting that β1-integrin is dispensable for AC tract fasciculation. Only *Emx1-Cre;Itgb1^fl/fl^* mutants phenocopy the Cortical Bundle phenotype observed in *Cas TcKO* mutants (B", white asterisks), indicating β1-integrin similarly functions in a population of cortical non-neuronal cells during TCA projection. (E) Quantification of the number of Cortical Bundles in *Emx1-Cre;Itgb1^fl/fl^* mutants and controls. Unpaired *t*-test, ****p < 0.0001, n = 6 for *Emx1-Cre;Itgb1^fl/fl^,* n = 5 for controls. Values provided are mean ± SEM. Scale bar = 250µm.

by our mutants here could be the displacement of subplate and other deep-layer neurons in *Emx1-Cre;TcKO* and *Emx1-Cre;Itgb1^fl/fl^* mutants.

Thus, we performed immunohistochemistry in *Emx1-Cre;TcKO* and *Emx1-Cre;Itgb1^fl/fl^* mutants at E16.5 (Fig 8) and P0 (Fig 9), using the markers NCAM-L1 and Tbr1. Tbr1 is a transcription factor known to regulate the specification of the subplate and layer VI cortical layers [76]. Indeed, we found that at E16.5 and P0, both *Emx1-Cre;TcKO* and *Emx1-Cre;Itgb1^fl/fl^*

PLOS Genetics

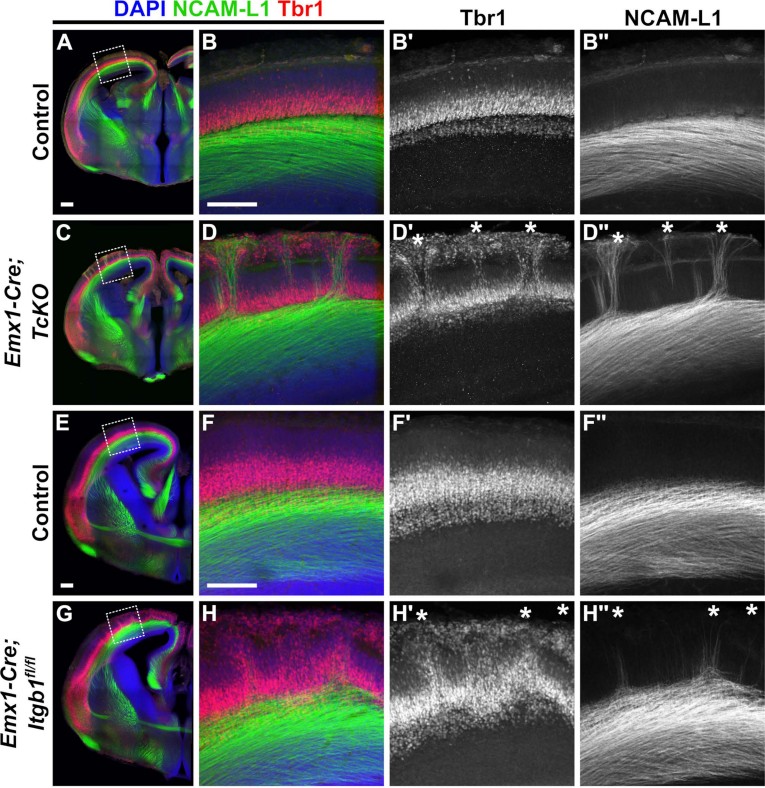

**Fig 8. Cortical Bundle phenotype in E16.5 *Itgb1* and *Cas TcKO* mutants correlates with displaced subplate cells.** (A-H") Immunohistochemistry of Tbr1 (red) and NCAM-L1 (green) in E16.5 *Emx1-Cre;TcKO* and *Emx1-Cre;Itgb1^{fl/fl}* mutants. Analysis reveals that severe cortical dysplasia is associated with Cortical Bundles. In both classes of conditional mutants, Tbr1+deep-layer/subplate cells are displaced to more superficial layers (D' and H', white asterisks). Cortical Bundles correlate with superficially displaced Tbr1+deep-layer/subplate cells (D" and H", white asterisks). 100% penetrance and expressivity. n=6 for *Emx1-Cre;TcKO*, n=6 for controls; n=8 for *Emx1-Cre;Itgb1,* n=8 for controls. Scale bar=250μm.

mutants displayed cortical dysplasia, as previously described [43]: lamination of deep-layer Tbr1+cells is highly disrupted, with columns of Tbr1+cells reaching the marginal zone (Figs 8A–B', 8C-D', 8E-F', 8G-H', 9A-B', 9C-D', 9E-F' and 9G-H'). Interestingly, in both *Emx1-Cre;TcKO* and *Emx1-Cre;Itgb1^{fl/fl}* mutants, 100% of the NCAM-L1+CBs associated with these displaced Tbr1+cells, at times with clear "tunnels" of Tbr1+cells containing CBs invading the marginal zone (Fig 8D–8D" and Fig 8H–8H", white asterisks; Fig 9D–9D" and Fig 9H–9H", white asterisks; n=5–8 per group, see figure legends for details). We have previously demonstrated that there are no cortical lamination defects in P7 *Nex-Cre;TcKO* and *Nex-Cre;Itgb1^{fl/fl}* mutants [43]. Additional analysis of *Nex-Cre;TcKO* animals at P0, using the markers NCAM-L1 and Tbr1, showed no apparent differences in cortical lamination between *Nex-Cre;TcKO* mutants and littermate controls (n=4 for *Nex-Cre;TcKO* mutants, n=4 for controls) (S10 Fig). Taken together, these results demonstrate that the disorganization of the subplate and deep cortical layers observed in pan-cortical *Cas TcKO* and *Itgb1* mutant mice is non-neuronal autonomous and strongly correlates with the ectopic CB phenotype.

## Discussion

In this study, we used conditional mouse genetics to assess the cortical-autonomous and neuronal-autonomous roles of *Cas* genes during forebrain white matter tract formation. Using a *Cas* triple conditional knockout (*Cas TcKO*) model, we provide strong genetic evidence that Cas proteins are required in two manners: 1) cortical- and neuronal-autonomously

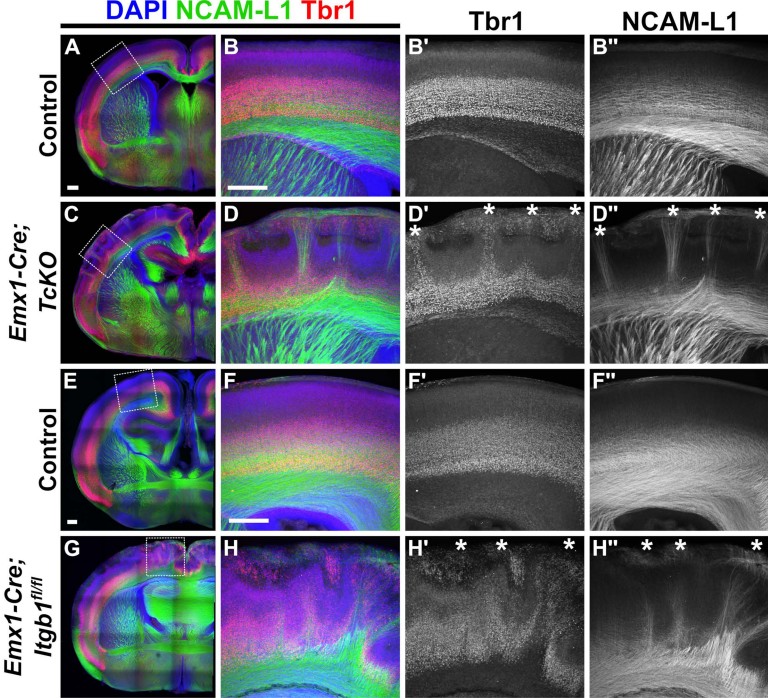

**Fig 9. Cortical Bundle phenotype in P0 *Itgb1* and *Cas TcKO* neonate mutants correlates with displaced subplate cells.** (A-H") Immunohisto-logical analysis of Tbr1 (red) and NCAM-L1 (green) in P0 *Emx1-Cre;TcKO* and *Emx1-Cre;Itgb1^{fl/fl}* mutants. Severe cortical dysplasia is associated with Cortical Bundles. In both classes of conditional mutants, Tbr1 + deep-layer/subplate cells are displaced to more superficial positions (D' and H', white asterisks). Cortical Bundles correlate with superficially displaced Tbr1 + deep-layer/subplate cells (D" and H", white asterisks). 100% penetrance and expressivity. n = 5 for *Emx1-Cre;TcKO*, n = 7 for controls; n = 6 for *Emx1-Cre;Itgb1*, n = 5 for controls. Scale bar = 250μm.

for proper fasciculation of the posterior branch of the Anterior Commissure (pAC) ([Figs 3B](), [3D]() and [5]()), and 2) cortical- and non-neuronal autonomously for thalamocortical axon (TCA) guidance ([Figs 3B"](), [3H]() and [5]()).

The AC is a highly evolutionarily conserved forebrain tract that, in mice, makes reciprocal connections between the Anterior Olfactory Nuclei of each olfactory bulb (via the anterior AC tract, aAC) and mediates reciprocal connections between the piriform and entorhinal cortices of the two hemispheres (via the posterior AC tract, pAC) [32]. Recent developmental studies have identified a small group of pioneering axons with unusual elongation kinetics that advance towards the embryonic midline at E13, preceding the majority of pAC axons that arrive later at E15 [33]. These late-arriving processes join the stalled pioneers to form a primary fascicle that crosses the embryonic midline in unison at E16 then innervates contralaterally and achieves specific target selection at E17 and E18, respectively [33]. In our study, we find that *Cas TcKO* mutants display defasciculation and ventral targeting of axons of the pAC tract ([Figs 3B'](), [3D]() and [5]()). However, these defects are only observable at later embryonic (E18.5) and postnatal (P0) stages ([Figs 4F]() and [3B](), respectively). Considering the timeline of pAC formation, this suggests that the axons that fail to fasciculate properly are a population of late "follower" axons arriving at the AC later in development.

Interestingly, we find that the defasciculating AC processes originate from the dorsolateral cortex, as they are labeled by lipophilic tracing from the somatosensory S2 cortex in all three classes of *Cas TcKO* mutants analyzed ([Figs 3D](); [5D]() and [5J]()). These results are somewhat unexpected: the pAC has been primarily described as carrying axons from ventrolateral cortical areas in mice [32]. The observation of DiI-labeled AC axons in both control and mutant animals after placement of dye crystals in S2 might suggest that a population of cortical axons projecting along the EC contributes to the formation of the AC. However, we cannot exclude the possibility that DiI cross-labeled other AC-projecting cortical axons

ventral to S2. We genetically confirmed the cortical origin of the defasciculating AC axons using the *Ai14 Cre*-dependent reporter in both the *Emx1-Cre* and *Nex-Cre* lines (Fig 6). Additionally, we observed intercalation of defasciculating tdTomato⁺ AC processes with other tdTomato⁻/NCAM-L1⁺ processes in the subpallium. This may suggest that these defasciculating axons may be failing to properly regulate homotypic versus heterotypic associations [77]. Taken together, these data support a neuronal-autonomous requirement of *Cas* genes for the guidance of a subset of cortical axons, which in the absence of these genes, mis-project into the ventral subpallium.

During canonical adhesion signaling, Cas proteins are recruited to Integrin Adhesion Complexes (IACs) following integrin receptor activation [34,39,78]. The integrins are a large family of transmembrane receptors that include 18 alpha (α) and 8 beta (β) subunits and are known to form 24 combinations of αβ heterodimers [79,80]. The β1-integrin subunit has been particularly well studied and is found in 12 of the known heterodimer combinations, mediating many intracellular signaling functions during CNS development [42,66–69,75,80,81]. We therefore analyzed the structure of the AC in *Itgb1* mutant neonates, which are deficient for the β1-integrin subunit. Surprisingly, conditional deletion of *Itgb1* using *Emx1-Cre* or *Nex-Cre* failed to phenocopy the AC defasciculation defects observed in *Cas TcKO* mutants (Fig 7), suggesting that β1-integrin is dispensable for proper fasciculation of this tract. This raises two intriguing possibilities: 1) there is compensation by another β subunit in *Itgb1*-deficient mice, or 2) Cas proteins are signaling downstream of a different type of receptor.

In support of the first possibility, β5-integrin, β6-integrin, and β8-integrin are expressed in the embryonic cortex (S7–S9 Figs) [75]. Thus, they might be acting as part of the adhesion receptors driving the fasciculation of the AC axons, or might compensate for the loss of *Itgb1* in mutant mice via the formation of heterodimers with α-integrins. For example, heterodimer formation between αV-integrin and β1-, β5-, β6-, and β8-integrin has been previously documented [80]. Of all α-subunits, αV-integrin appears as a prime candidate to modulate this process: in addition to being able to form heterodimers with multiple β-integrins, αV-integrin is known to have a critical role in neurodevelopment and is expressed in axons of white matter tracts in the cerebral cortex [82,83]. Moreover, αV-integrin-containing dimers bind to the ECM molecules laminin, fibronectin, vitronectin, tenascin, and collagen [68,75]. Importantly, laminin, fibronectin, and tenascin are expressed by a population of midline glial cells that help coordinate the development of the pAC [29]. One possible scenario then is that β5-integrin, β6-integrin, or β8-integrin is acting as a functional partner with αV-integrin during AC fasciculation, or can at least compensate for the loss of *Itgb1.*

The second possible explanation for the failure of *Itgb1* mutants to phenocopy the AC defasciculation defects presented by *Cas TcKO* animals is that Cas proteins are signaling downstream of an entirely different receptor system. It has previously been shown that phospho-activation of Cas can be modulated by CXCL12/CXCR4 [44], Neuropilin [45], Netrin [46], and Eph/Ephrin [84] signaling. Moreover, mutations in several genes that participate in these signaling pathways or have known roles in adhesion have been shown to affect the development of the AC; however, most of these mutations result in thinning or absence of the AC tracts, not defasciculation of AC axons [85–107].

Notably, *Ephb2/Nuk* mutants [86] and *Ephb1/Ephb2* double-mutants [87] display defasciculation and ventral targeting of pAC axons, which partially phenocopy *Cas TcKO* mutants. Moreover, biochemical studies have implicated cross-talk between β1-integrin and EphB/EphrinB pathways in various cell types [108–111], including retinal ganglion cells [112] and mossy fiber axons [113]. It is possible that these signaling cascades may be coordinating phosphorylation of Cas proteins in cortical neuron axons as well. Determining whether Cas proteins are working downstream of Eph/Ephrin systems during AC projection and fasciculation will be an interesting goal for future studies.

The second white matter tract phenotype presented by *Cas TcKO* mutants is the appearance of large ectopic bundles of axons mis-projecting towards the pial surface of the cortex (Fig 3A" and 3B"), which we termed Cortical Bundles (CBs). Lipophilic tracing experiments revealed the origin of these mis-projections to be thalamic (Fig 3F and 3H). Analysis of cortical bundle formation in *Emx1-Cre;TcKO* revealed a comparable phenotype to that observed in *Nestin-Cre;TcKO*, strongly suggesting a cortical-autonomous role for *Cas* genes in CB formation. Further genetic experiments demonstrated

that this phenotype is non-neuronal-autonomous: conditional deletion of *Cas* genes using *Nex-Cre* failed to recapitulate the CB phenotype (Fig 5H" and 5L). From this, we inferred that *Cas* gene function is required by some non-neuronal cortical-resident population responsible for coordinating TCA entry. Similarly, analysis of *Itgb1* mutants revealed that conditional deletion using *Nex-Cre* did not result in CB formation (Fig 7D"), indicating a comparable cortical-autonomous, non-neuronal-autonomous role for β1-integrin. However, we cannot formally exclude a thalamus-autonomous role for *Itgb1* or *Cas* genes in TCA pathfinding, as we did not ablate these genes exclusively in the thalamus. Future studies using thalamus-specific Cre drivers would be needed to definitively address this question.

During normal development, TCA innervation of the cortex is regulated by a transient structure known as the subplate, which acts as an intermediate target for TCAs until their target layer (layer IV) is fully formed [15]. Arriving TCA projections stall in the subplate for several days and grow extensive side processes within the subplate during this waiting period [18–20]. Upon entering the cortical plate, TCAs form transient microcircuits with subplate neurons and maturing cortical layer IV neurons, a process which appears to facilitate the proper targeting of layer IV and thus the organization of mature circuits [23,114,115]. Mutations that disrupt the laminar organization of the subplate, such as *reeler* mutants [116] and *Cdk5* mutants [117], result in aberrant TCA projection patterns within the cortical plate, speaking to the developmental importance of the subplate in TCA guidance. Previous work from our lab and other labs has demonstrated that *Cas TcKO* and *Itgb1* mutants have severely disrupted laminar organization of the cortex caused by breakages in the basal lamina that destabilize radial glial endfeet attachment, resulting in improper migration of cortical neurons and severe cortical dysplasia [43,74,118].

Taking all of this into consideration, we asked whether the disruption of subplate and layer VI organization observed in *Cas TcKO* and *Itgb1* mutants could be a contributing developmental factor to the cortical-autonomous defects we observed in TCA guidance. Using the subplate/Layer VI marker Tbr1, we observed severe displacement of the deep cortical layers at E16.5 (Fig 8) and P0 (Fig 9) in both *Cas TcKO* and *Itgb1* mutants. Co-labeling with NCAM-L1 revealed that 100% of ectopic CB projections correlate with sites of displaced Tbr1+cells (Figs 8D and 8H; 9D and 9H).

Within the context of our previous work [43], these results suggest a model whereby *Cas* genes are required in cortical radial glial cells to regulate endfeet attachment, stabilization of the basement membrane, and glial process organization, which subsequently aid in neuronal migration and formation of the cortical layers, including the subplate. In the absence of *Cas* or *Itgb1* genes, these radial glial tracks are disrupted, resulting in mispositioned subplate neurons (Figs 8D and 8H; 9D and 9H) [43]. These mispositioned subplate neurons could potentially recruit TCAs for innervation, causing ectopic cortical bundles to innervate the proper intermediate target positioned in the wrong location. Hence, our model (S4 Fig) is that while the role of *Cas* genes is non-neuronal-autonomous (acting in radial glia rather than in the neurons themselves), the mispositioned Tbr1+neurons in *Emx1-Cre;TcKO* mutants drive the TCA misprojection phenotype. These results, however indirect, add to an extensive body of literature suggesting that proper lamination and formation of the subplate is required for appropriate TCA afferent projection [15–19,21–23,61,119–124].

Collectively, this work establishes Cas proteins as essential mediators of forebrain axon tract development and provides genetic mechanistic insights into their cell-type-specific functions during cortical lamination-dependent thalamo-cortical guidance and neuronal-autonomous commissural fasciculation. We build on previous genetic evidence that *Cas* genes function radial glial-autonomously downstream of the β1-integrin receptor to ensure proper lamination of the cortex [43], demonstrating that errors in lamination result in disruptions to the subplate. Based on our data, we propose that this misplacement of the subplate might cause aberrant thalamocortical axon guidance and the formation of ectopic bundles that mis-project towards the pial surface. Further, we provide genetic evidence that *Cas* genes function independently of the β1-integrin receptor in a neuronal-autonomous manner during fasciculation and guidance of the Anterior Commissure, demonstrating that a subset of late-arriving axons fail to properly fasciculate with the posterior branch of the Anterior Commissure. These defasciculating axons display improper ventral targeting upon crossing the pallial-subpallial boundary. Our findings add to a growing body of evidence suggesting essential roles for Cas proteins in adhesion signaling during axon guidance and migration [34,41–43,46,52,125,126].

## Materials & methods

**Ethics approval:** All animal procedures presented here were performed according to the University of California, Riverside's Institutional Animal Care and Use Committee (IACUC)-approved guidelines, under protocol A-8.

**Animals and Genotyping:** For embryonic collections, the morning of vaginal plug observation was designated as embryonic day 0.5 (E0.5) and the day of birth was postnatal day 0 (P0). The generation of the *Cas TcKO* model has been previously described [42,127,128]. Swiss Webster mice were purchased from Taconic Biosciences (Model: SW-F and SW-M). The following lines were purchased from The Jackson Laboratory: *Nestin-Cre* (Stock: 003771), *Emx1-Cre* (Stock: 005628), *Ai14/R26^{LSL-tdTomato}* (Stock: 007914), and *Itgb1^{fl/fl}* (Stock: 004605). Genotyping for *Nestin-Cre* was carried out using the following primers: F: CCT TCC TGA AGC AGT AGA GCA R: GCC TTA TTG TGG AAG GAC TG. Genotyping for *Emx1-Cre* was carried out using the following primers: F: CCA TAT CAA CCG GTG GCG CAT C and R: TCG ATA AGC TTG GAT CCG GAG AG. Genotyping for *Ai14/R26^{LSL-tdTomato}* was carried out using the following primers: WTF: AAG GGA GCT GCA GTG GAG TA, WTR: CCG AAA ATC TGT GGG AAG TC, MutF: CTG TTC CTG TAC GGC ATG G, and MutR: GGC ATT AAA GCA GCG TAT CC. Genotyping for *Itgb1^{fl/fl}* mice was carried out using the primers: F: CGG CTC AAA GCA GAG TGT CAG TC and R: CCA CAA CTT TCC CAG TTA GCT CTC. *Nex-Cre* mice were kindly provided by Drs. Nave and Goebbels [64,129]. Genotyping for *Nex-Cre* was carried out using the following primers: Primer 4: GAG TCC TGG AAT CAG TCT TTT TC, Primer 5: AGA ATG TGG AGT AGG TGA C, and Primer 6: CCG CAT AAC CAG TGA AAC AG. Tail biopsies were digested with Quanta tail extraction reagent kit (cat: 95091–025). PCR reactions were prepared using GoTaq Master Mix (Promega cat: PRM7123). All animal procedures were performed according to the University of California, Riverside's Institutional Animal Care and Use Committee (IACUC) guidelines. All procedures were approved by UC Riverside IACUC.

**Immunohistochemistry:** Embryos, neonates, and adult mice were transcardially perfused with ice-cold 1xPBS and subsequently dissected in cold 1xPBS. Brain tissue was post-fixed in 4% PFA (diluted in 1xPBS) for 2hrs – overnight at 4°C. Tissue samples were processed immediately as floating sections or cryopreserved in a solution of 30% sucrose in 1xPBS before embedding in Tissue Plus O.C.T. Compound (Fisher HealthCare, cat: 4585) for long-term storage. Tissue processed as floating sections was embedded in 3% agarose and sectioned coronally as 100–150µm sections on a vibratome (Lecia, cat: VT1000S). Floating sections were incubated with a permeabilization solution containing 3% bovine serum albumin (BSA) and 0.3% TritonX-100 in 1xPBS for 4hrs – overnight at 4°C. Primary and secondary antibody mixes were diluted in a solution of 5% goat serum made with permeabilization solution. All antibody incubations were overnight at 4°C with gentle agitation. Cryoprotected samples were sectioned coronally at 25–50µm on a cryostat (Leica cat: CM3050). Cryosections were dried for 20 minutes at room temperature before blocking in freshly prepared blocking buffer (10% goat serum and 0.1% TritonX-100 in 1xPBS) for 1hr before primary antibody addition. Antibodies were then prepared in blocking buffer with reduced goat serum (5%) and incubated in a humidified chamber overnight at 4°C. The antibodies and concentrations used in this study: rabbit anti-p130Cas (E1L9H) (1:200, Cell Signaling Technology cat: 13846), rat anti-NCAM-L1 (1:500, Millipore cat: MAB5272), chicken anti-Nestin (1:500, Aves cat: NES), chicken anti-GFP (1:500, Aves cat: GFP-1020), rabbit anti-dsRed (1:500, Takara Bioscience cat: 632496), rabbit anti-Tbr1 (1:500, Abcam cat: ab31940 [polyclonal]), rabbit anti-Tbr1 (1:500, Abcam cat: ab183032 [monoclonal]), mouse anti-CD29 (Integrin beta 1) (1:100, eBioscience cat: 14-0299-82), and rabbit anti-Integrin β1 (D6S1W) (1:100, Cell Signaling Technology cat: 34971). Secondary antibodies were purchased from ThermoFisher and diluted to 1:1000 from the stock concentration: goat anti-chicken 488 (cat: A-11039), goat anti-chicken 546 (cat: A-11040), goat anti-rabbit 488 (cat: A-11034), goat anti-rabbit 546 (cat: A-11035), goat anti-rabbit 647 (cat: A-21244), goat anti-rat 488 (cat: A-21208), goat anti-rat 546 (cat: A-11081), and goat anti-mouse 647 (cat: A-21236). Nuclear counterstain was achieved by co-incubation with 1µg/ml DAPI (ThermoFisher cat: 50850585) in all secondary antibody incubations. Tissue was mounted using Fluoro Gel with DABCO (Electron Microscopy Sciences, cat: 17985–02). Confocal scanning images were acquired on a Leica DMi8 using a 10x objective (NA=0.40) or 20x objective (NA=0.75), and z-stacks were acquired with 20–40 steps of sizes of 2–4µm.

**Fluorescent *in situ* Hybridization:** Embryos were perfused and dissected as described above for immunohistochemistry. Brain tissue was then post-fixed in 4% PFA (in 1xPBS) overnight at 4°C and cryopreserved as previously described.

Fluorescent *in situ* hybridization was performed on 15 µm cryosections from E14.5, E16.5, or E18.5 embryos using the RNAScope Multiplex Fluorescent Detection Kit v2 from Advanced Cell Diagnostics according to the manufacturer's instructions (ACD, 323110) [130]. Briefly, ssDNA "z-probes" hybridize complementary to the RNA of interest. Oligos bind to the tail region of the z-probe and are bound to amplifiers labeled with horseradish peroxidase (HRP) and fluorophores. *Itgb5* (ACD, 404311), *Itgb6* (ACD, 312501), and *Itgb8* (ACD, 407931) probes generated by Advanced Cell Diagnostics were uniquely amplified with Opal dyes (Akoya Biosciences, FP1487001KT, FP1488001KT, FP1496001KT) and counterstained with DAPI (1 µg/ml). Slides were mounted using Fluoro Gel with DABCO. Confocal scanning images were acquired on a Leica DMi8.

**Axonal Tracing:** Lipophilic tracing of white matter tracts was performed by manually placing small crystals of either DiA (Invitrogen cat: D3883) or DiI (Invitrogen cat: D282) into neonate tissue dissected as described above for immunohistochemistry and postfixed in 4% PFA (in 1xPBS) overnight at 4°C. DiA crystals were placed in the Somatosensory 2 (S2) region of the cortex, while DiI crystals were placed in the laterodorsal thalamus. Lipophilic crystals were allowed to diffuse for 8 weeks at 37°C in a solution of 4% PFA in 1xPBS, with the solution changed daily for the first 4 days and weekly thereafter. Tissue was embedded in 3% agarose in 1xPBS and sectioned as 150 µm sections on a vibratome (Lecia, cat: VT1000S). Sections were imaged freshly sliced on a Zeiss Axio Imager 2 microscope.

**Quantification of Stray Axon Bundles and Cortical Bundles:** Quantification of stray axon bundles defasciculating from the EC-AC tract and mis-projecting Cortical Bundles (CBs) was done using ImageJ. The number of stray axon bundles in the subpallial area and the number of CBs in the cortex were counted manually for each section per animal using the ImageJ multipoint tool. Three to twelve independent animals were analyzed per group, depending on the experiment, and two to twelve sections were analyzed per animal. Bar graphs represent the group mean ± standard error of group mean, with plotted values representing the mean from each animal analyzed. The datasets were then tested for normality using the Shapiro–Wilk test and QQ plot. For normally distributed datasets, Unpaired *t*-test was performed for comparing two independent samples and two-way ANOVA, followed by Tukey honestly significant difference (HSD) post hoc test, was used for multiple comparisons.

## Supporting information

**S1 Table. Numerical values for graphs in Figs 3–7.** The number of stray axon bundles in the subpallial area and the number of CBs in the cortex for each section from each animal analyzed. Data are organized by figure: Fig 3 (Sheet 1), Fig 4 (Sheet 2), Fig 5 (Sheet 3), Fig 6 (Sheet 4), and Fig 7 (Sheet 5).
(XLSX)

**S1 Fig. p130Cas protein is expressed in the developing neocortex.** (A-F') Immunohistochemistry showing the developmental expression pattern of p130Cas protein (green) during wild-type cortical development. At E14.5, p130Cas protein expression overlapped with the radial glial cell marker Nestin (red) in the cortical plate and at the pial surface (B-B'). At E16.5, p130Cas expression is strongest in the cortical white matter, subventricular zone, and marginal zone (D-D'). By E18.5, p130Cas is robustly expressed in the subventricular and intermediate zones (F-F'). Weaker expression of p130Cas is also detected at the marginal zone and pial surface at E16.5 (D') and E18.5 (F'), overlapping with Nestin expression (D and F). MZ = Marginal Zone; CP = Cortical Plate; IZ = Intermediate Zone; SVZ = Subventricular Zone; VZ = Ventricular Zone. Scale bar = 250µm (A, C-F'). Scale bar = 100µm (B-B'). E14.5: n = 4; E16.5: n = 3; and E18.5: n = 4.
(TIF)

**S2 Fig. The p130Cas antibody is specific for p130Cas protein.** (A-H') Immunohistochemistry of E16.5 *Nes-Cre;TcKO* and littermate control embryos (*CasL$^{-/-}$;Sin$^{-/-}$;p130Cas$^{fl/+}$*) demonstrates p130Cas signal (green) is not detected in *Nes-Cre;TcKO* mice. Control embryos strongly express p130Cas protein in the developing neocortex (D') and cortical white matter tracts (A'), including the External Capsule (EC) and Anterior Commissure (AC) (B'). p130Cas signal is lost in *Nes-Cre;TcKO* embryos (E',

F', and H') except for in the hypothalamic region (E' and G', white asterisk). Note that the expression of p130Cas looks nearly identical in control *CasL⁻/⁻;Sin⁻/⁻;p130Cas^{fl/+}* embryos and wild-type embryos (Figs 1 and S1). MZ = Marginal Zone; CP = Cortical Plate; IZ = Intermediate Zone; SVZ = Subventricular Zone; VZ = Ventricular Zone. Scale bar = 250μm (A-A', C-C', E-E', and G-G'). Scale bar = 100μm (B-B', D-D', F-F', and H-H'). n = 3 animals for *Nes-Cre;TcKO* and control.
(TIF)

**S3 Fig. Placement of DiA and DiI crystals in *Nes-Cre;TcKO* mutants and controls.** (A-F) DiA and DiI crystal placement in coronal sections from P0 *Nes-Cre;TcKO* and control mice. DiA crystals were placed into the Somatosensory 2 (S2) region of the cortex (C), while DiI crystals were placed along the central region of the thalamus (F). Scale bar = 250μm.
(TIF)

**S4 Fig. Illustration of cell-type-specific requirements of *Cas* gene function for TCA pathfinding.** (A-C) Representative model depicting the differences in *Cas* gene expression in Control, *Emx1-Cre;TcKO*, and *Nex-Cre;TcKO* animals. (A) In control animals, *Cas* genes are expressed in radial glial cells (RGCs; solid blue), postmitotic immature migrating neurons (solid yellow), and post-migratory excitatory cortical neurons (solid grey), including the subplate (solid red). (B) *Emx1-Cre* drives recombination in cortical RGCs. Thus, in *Emx1-Cre;TcKO* mutants, *Cas* genes are ablated in RGCs (hollow blue), immature migrating excitatory neurons (hollow yellow), and excitatory neurons (hollow grey), including the subplate (hollow red). Thalamocortical axons (TCAs) project to mispositioned subplate cells, resulting in the Cortical Bundle (CB) phenotype observed in *Emx1-Cre;TcKO* mutant mice. (C) In *Nex-Cre;TcKO* animals, *Nex-Cre* drives recombination in early postmitotic excitatory cortical neurons. *Cas* genes are therefore present in RGCs (solid blue), but not in migrating neurons (hollow yellow) and post-migratory neurons of the developing neocortex (hollow grey and hollow red). TCA projections are normal in *Nex-Cre;TcKO* mutants. ECM = Extracellular Matrix; MZ = Marginal Zone; CP = Cortical Plate; VZ = Ventricular Zone; RGC = Radial Glial Cell; TCA = Thalamocortical Axons. Created with BioRender.com.
(TIF)

**S5 Fig. Rabbit anti-β1-integrin antibody is not specific for β1-integrin protein.** (A-F') Immunohistochemistry of E16.5 *Emx1-Cre;Itgb1* and littermate control animals using the rabbit-anti-β1-integrin antibody (Cell Signaling Technology) demonstrates that putative β1-integrin signal (green) is still present in the cortex of *Emx1-Cre;Itgb1* mice. Control embryos show signal in the marginal zone (C'), cortical plate, and all major cortical white matter tracts (A'), including thalamocortical axons (C'), the External Capsule (EC) (E'), and the Anterior Commissure (AC) (E'). This signal is still observed in *Emx1-Cre;Itgb1* animals, including the cortical plate, EC, and AC (B', D', and F'). Scale bar = 250μm (A-B'). Scale bar = 100μm (C-F'). n = 4 animals for *Emx1-Cre;Itgb1* and control.
(TIF)

**S6 Fig. Mouse anti-β1-integrin antibody is not specific for β1-integrin protein.** (A-F') Immunohistochemistry of E16.5 *Emx1-Cre;Itgb1* and littermate control animals using the mouse-anti-β1-integrin antibody (eBioscience) shows that putative β1-integrin signal (green) is still present in the cortex of *Emx1-Cre;Itgb1* mice. Control embryos show signal in the vasculature (C'), the piriform cortex (E'), marginal zone, and all major cortical white matter tracts (A'), including thalamocortical axons (C'), the External Capsule (EC) (E'), and the Anterior Commissure (AC) (E'). Strong signal is still detected in the cortex, AC, and EC of *Emx1-Cre;Itgb1* animals (B'). High-magnification images show clear β1-integrin signal in thalamocortical axons (D') and the EC and AC (F'). Scale bar = 250μm (A-B'). Scale bar = 100μm (C-F'). n = 4 animals for *Emx1-Cre;Itgb1* and control.
(TIF)

**S7 Fig. *Itgb5, Itgb6*, and *Itgb8* are expressed in the neocortex at E14.5.** (A-D''') RNAScope for *Itgb5* (yellow), *Itgb6* (magenta), and *Itgb8* (cyan) in the developing neocortex at E14.5. *Itgb6* is expressed diffusely throughout the developing

neocortex (A″, B″, and C″), while *Itgb8* is robustly expressed in the subventricular and ventricular zones of the Primary Somatosensory Cortex (S1) (B‴) and Secondary Somatosensory Cortex (S2) (C‴). *Itgb5* appears to be weakly expressed in the ventricular zone as well as in the vasculature (A'). High magnification images show diffuse expression of *Itgb6* (D″) and *Itgb8* (D‴) in the Piriform Cortex (P). S1 = Primary Somatosensory Cortex; S2 = Secondary Somatosensory Cortex; P = Piriform Cortex. MZ = Marginal Zone; CP = Cortical Plate; IZ = Intermediate Zone; SVZ = Subventricular Zone; VZ = Ventricular Zone. Scale bar = 250µm (A-A‴). Scale bar = 100µm (B-D‴). n = 3 animals.
(TIF)

**S8 Fig.  *Itgb5*, *Itgb6*, and *Itgb8* are expressed in the neocortex at E16.5.** (A-D‴) RNAScope for *Itgb5* (yellow), *Itgb6* (magenta), and *Itgb8* (cyan) in the developing neocortex at E16.5. *Itgb5*, *Itgb6*, and *Itgb8* are all expressed in the ventricular zone of the Primary Somatosensory Cortex (S1) (B'-B‴) and Secondary Somatosensory Cortex (S2) (C'-C‴). *Itgb5* also appears to be expressed in the vasculature (A'). Weak expression of *Itgb6* (B″ and C″) and *Itgb8* (B‴ and C‴) is detected in the marginal zone of S1 and S2. Expression of *Itgb6* (D″) and *Itgb8* (D‴) is observed in the Piriform Cortex (P). S1 = Primary Somatosensory Cortex; S2 = Secondary Somatosensory Cortex; P = Piriform Cortex. MZ = Marginal Zone; CP = Cortical Plate; IZ = Intermediate Zone; SVZ = Subventricular Zone; VZ = Ventricular Zone. Scale bar = 250µm (A-A‴). Scale bar = 100µm (B-D‴). n = 3 animals.
(TIF)

**S9 Fig.  *Itgb5*, *Itgb6*, and *Itgb8* are expressed in the neocortex at E18.5.** (A-D‴) RNAScope for *Itgb5* (yellow), *Itgb6* (magenta), and *Itgb8* (cyan) in the developing neocortex at E18.5. All three transcripts are strongly expressed in the ventricular zone of the Primary Somatosensory Cortex (S1) (B'-B‴). *Itgb5* appears to be weakly expressed in the vasculature (B', C', and D'). *Itgb8* is diffusely expressed in the Secondary Somatosensory Cortex (S2) (C‴). Expression of *Itgb6* (D″) and *Itgb8* (D‴) is detected in the Piriform Cortex (P). S1 = Primary Somatosensory Cortex; S2 = Secondary Somatosensory Cortex; P = Piriform Cortex. MZ = Marginal Zone; CP = Cortical Plate; IZ = Intermediate Zone; SVZ = Subventricular Zone; VZ = Ventricular Zone. Scale bar = 250µm (A-A‴). Scale bar = 100µm (B-D‴). n = 3 animals.
(TIF)

**S10 Fig.  Positioning of deep-layer neurons is normal in P0 *Nex-Cre;TcKO* animals.** (A-D″) Immunohistochemistry of P0 *Nex-Cre;TcKO* animals using the axonal marker NCAM-L1 (green) and the deep layer marker Tbr1 (red) shows no apparent differences in lamination between *Nex-Cre;TcKO* and control cortices. Normal positioning of deep-layer/subplate cells is observed in *Nex-Cre;TcKO* animals (D'). This correlated with the normal fasciculation and pathfinding of NCAM-L1+ (green) cortical white matter tracts. n = 4 for *Nex-Cre;TcKO* and controls. Scale bar = 250µm (A and C). Scale bar = 100µm (B-B″, D-D″).
(TIF)

## Acknowledgments

We would like to thank Dr. Teresa Ubina for critically reading the manuscript and providing helpful comments. We would like to thank Drs. Nave and Goebbels for the *Nex-Cre* mice. We would also like to thank Drs. Sachiko Seo and Mineo Kurokawa, and Dr. Konstantina Alexandropoulos for sharing the *CasL⁻ᐟ⁻* and *Sin⁻ᐟ⁻* mouse lines, respectively.

## Author contributions

**Conceptualization:** Martin M. Riccomagno.

**Funding acquisition:** Martin M. Riccomagno.

**Investigation:** Jason A. Estep, Alyssa M. Treptow, Payton A. Rao, Patrick Williamson, Wenny Wong.

**Methodology:** Jason A. Estep, Alyssa M. Treptow, Payton A. Rao, Patrick Williamson, Wenny Wong, Martin M. Riccomagno.

**Writing – original draft:** Jason A. Estep, Alyssa M. Treptow, Martin M. Riccomagno.

**Writing – review & editing:** Jason A. Estep, Alyssa M. Treptow, Payton A. Rao, Martin M. Riccomagno.

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
