## [Decision Letter · Decision Letter 0]

4 Aug 2025

PGENETICS-D-25-00757

Functional Role for Cas Cytoplasmic Adaptor Proteins During Cortical Axon Pathfinding

PLOS Genetics

Dear Dr. Riccomagno,

Thank you for submitting your manuscript to PLOS Genetics. After careful consideration, we feel that it has significant merit but does not yet fully meet PLOS Genetics's publication criteria as it currently stands. Therefore, we invite you to submit a revised version of the manuscript that addresses the points raised during the review process.

Please submit your revised manuscript within 30 days Sep 03 2025 11:59PM. If you will need more time than this to complete your revisions, please reply to this message or contact the journal office at plosgenetics@plos.org. Please include the following items when submitting your revised manuscript:

We look forward to receiving your revised manuscript.

Kind regards,

Ophir Klein

Academic Editor

PLOS Genetics

Fengwei Yu

Section Editor

PLOS Genetics

Aimée Dudley

Editor-in-Chief

PLOS Genetics

Anne Goriely

Editor-in-Chief

PLOS Genetics

**Journal Requirements:**

https://journals.plos.org/plosgenetics/s/submission-guidelines#loc-parts-of-a-submission

4) Thank you for stating "The data that support the findings of this study are publicly available from https://figshare.com with the identifier(s) [TBD upon initial acceptance by PLOS Genetics]." We strongly recommend all authors decide on a data sharing plan before acceptance, as the process can be lengthy and hold up publication timelines. Please note that, though access restrictions are acceptable now, your entire data will need to be made freely accessible if your manuscript is accepted for publication. This policy applies to all data except where public deposition would breach compliance with the protocol approved by your research ethics board. If you are unable to adhere to our open data policy, please kindly revise your statement to explain your reasoning and we will seek the editor's input on an exemption. Please be assured that, once you have provided your new statement, the assessment of your exemption will not hold up the peer review process.

2) If any authors received a salary from any of your funders, please state which authors and which funders..

**Reviewers' comments:**

Reviewer's Responses to Questions

Reviewer #1: I agree with the Review Commons reviewers on all the positive aspects of the manuscript. This is a well designed and executed study that will be of specific interest to those studying axon guidance and of broad interest to those studying neural development and circuit formation. The data are convincing and support the conclusions, the figure presentation is easy to follow, and the manuscript is well written. The study demonstrates several novel and important findings for the field.

The authors have adequately addressed most critiques from the Review Commons reviews. The majority of the comments were addressed by including new or clarifying text in the Results, Methods, or Discussion. Critiques from Reviewer #2 regarding integrin family members were not addressed experimentally. However, in this reviewers opinion these are not necessary experiments as they are not integral to supporting the main conclusions of the manuscript and would be outside the scope of the current study.

A few very minor revisions would further strengthen the manuscript.

1) I fully agree with Reviewer #1 from the Review Commons reviews that the terminology of cortical-autonomous and non-neuronal autonomous is challenging. At best it is cumbersome, at worst it is sometimes inaccurate. I highly recommend finding new wording to describe these interpretations. A good example is in the paragraph comprising lines 267-274. The sentence "The proper fasciculation of AC axons appears to require Cas in a neuron-autonomous manner, likely reflecting a requirement for Cas genes in the projecting axons themselves." The second part of the sentence is much clearer than the first half. "Neuronal-autonomous" is unnecessary and can be confusing, especially since the authors point out that the mutant axons might be erroneously following non-mutant WT axons. Continuing in that paragraph: "In contrast, the proper projection of TCA afferent into the cortex appears to have a non-neuronal-autonomous requirement for Cas genes. As such, we infer that Cas genes are required by some non-neuronal cortical cell population..." The second sentence is much more effective in conveying the message than the first sentence. "Non-neuronal-autonomous" is not a phrase that is used often in the field, and is likely to confuse many people. It would be much clearer to say something like "required by a non-neuronal population, but not by neurons themselves." Also, to say that the bundling phenotype is "non-neuronal-autonomous" can be understood as inaccurate, since the displaced neurons are likely causing the bundling phenotype. It's true that the Cas genes seem to be required in the progenitors and not in the neurons, but the neurons are definitely involved. Because of the high likelihood that readers will be confused by these terms, I highly recommend completely removing the terms "cortical autonomous", "neuronal-autonomous", and non-neuronal-autonomous" and just using different words to describe these ideas.

2) Line 254: "(i.e. requiring Cas gene function in the cortical plate itself)" is not accurate. The interpretation is that Cas genes are required in the developing cortex (or developing neocortex or developing dorsal forebrain, etc), but not necessarily in the cortical PLATE (which refers to the postmitotic neurons that are not in the VZ/SVZ, IZ, MZ).

3) Line 290: Sentence ending with "n=3 for controls)" is missing a period at the end.

4) The term "interpolate" (e.g., lines 292, 305, etc) is a bit unusual. Consider changing to "intercalate". I understand they can mean the same thing, but it seems "intercalate" is used more frequently to describe axons and "interpolate" is used more frequently in the mathematical sense. However, if this is the preferred term in the field, feel free to ignore.

5) Lines 350-351: the phrase "(Figure 7 and Figure 8, respectively)" is in an unusual place in the sentence, making it unclear whether Figure 7 and Figure 8 are split based on ages (E16.5 vs P0), mutant genotype (TcKO vs Itgb1), or markers (NCAM-L1 vs Tbr1).

6) This is not a major point but may be worth thinking about and possibly discussing: The number of stray axons in mutant brains is always bimodal, with some around 5 and others around 10 - is this meaningful or interesting? The exception is when using the Ai14 reporter, where all are at 10 - is there a reason for this? In your other brains where the mutant axons are not traced, are you missing some stray axons to get the average of 5 in some brains?

Reviewer #2: Review comments for PLoS Genetics

Estep et al., Cas Cytoplasmic Adaptor Proteins & Axon Pathfinding

Estep et al., take a genetic approach to testing the role of Cas proteins in cortical axon pathfinding. They show specific deficits in the anterior commissure and thalamocortical axons by deleting three of the four Cas proteins with complementary Cre drivers. I agree the findings presented are supported by the data. The data are generally of high quality. I am reviewing this in concert with the comments from the Review Commons mechanism with the mission to determine if the revisions and plan in response to those comments are sufficient. I do believe that most of my concerns have been addressed in the previous reviews and I find the changes made, or planned changes, will be sufficient to produce a rigorous manuscript. The previous comments about the somewhat limited impact of the work are valid but that is an editorial concern beyond the purview of this reviewer. I have some further minor comments to try and improve this manuscript beyond the first set of comments.

1. The authors pursue a genetic strategy which will delete three of the four Cas proteins. There is no mention of how these three are chosen or why Cass4 is omitted. I am not suggesting a quadruple knockout is necessary but the rationale may be useful to share.

2. Fewer non-standard abbreviations would likely help readers not fully immersed in the field. These include PSPB, DTB, a/pAC, CB.

3. The colocalization in the cortical plate at E14.5 in Fig 1 (p130Cas & NCAM) is not apparent. Other data points are convincing.

4. The authors use the common convention of abbreviating p-values with annotations like ****. I think a better and emerging style is to show the actual p-value and to let the reader make their own conclusions about statistical “significance.”

5. In figure 4b’, the Emx1-Cre defasciculation is a bit hard to see.

6. I personally do not prefer A, A’, A’’ vs. A,B,C in labeling figures but that is a matter of personal opinion.

7. Lines 320-335: can this be reorganized to tell the story in the same logical order used elsewhere in the manuscript?

8. The graphical abstract suggested in Review Commons would indeed be a nice addition.

9. The values plotted are the averages from each animal. This should be explicitly stated in the methods.

Reviewer #3: In this manuscript by Estep et al., the authors use conditional in vivo mouse genetics to study roles for Cas family intracellular adaptor proteins in forebrain axon tract development. They report two phenotypes after simultaneous nervous system-wide deletion of three Cas family genes – (1) defasciculation and misprojection of anterior commissure axons and (2) ectopic formation of thalamocortical axon bundles that penetrate the cortex. Further investigation using specific Cre lines and other conditional knockout alleles demonstrates that the anterior commissure defect results from a requirement for Cas genes in cortical projection neurons, whereas thalamocortical axons are misguided due to Cas functional requirements in cortical lamination, as ectopic axon bundles are confined to sites of disrupted cortical layer formation. Overall, this study uncovers novel functions for Cas family genes in forebrain axon organization, one of which likely reflects a direct role in axon guidance and/or fasciculation, while another one is indirect and based in the previously established role of Integrin-Cas signaling in radial glia organization and cortical neuron migration.

As stated during previous review, the data are of overall high quality, analyses are rigorous, and conclusions are well-supported by the data. The revision plan presented by the authors is solid and will provide additional data on Integrin expression, improve the overall presentation of data, and add schematics to clarify mechanistic interpretations.

Without identification of a novel signaling pathway operating upstream of Cas proteins in axon guidance/bundling the manuscript remains somewhat open-ended, yet the elegant and rigorous dissection of Cas function in forebrain axon tract formation through mouse genetics makes a significant contribution to the filed in itself. In their response to previous reviews, the authors highlight four drivers of novelty/significance that could be fleshed out in a more detailed discussion section. However, I do not agree that challenging "the assumption that Cas proteins always function downstream of beta1-Integrin" is all that novel, as multiple other ligand-receptor systems have been implicated upstream of Cas activation in neural circuit development. These pathways are mentioned multiple times in the manuscript and could be the focus of extended discussion, instead of listing many knockout mouse lines that exhibit anterior commissure defects, even if they do not resemble the Cas knockouts and there is no evidence implicating the relevant molecules upstream of Cas proteins.

**Have all data underlying the figures and results presented in the manuscript been provided?**

Reviewer #1: Yes

Reviewer #2: Yes

Reviewer #3: Yes

PLOS authors have the option to publish the peer review history of their article (what does this mean? ). If published, this will include your full peer review and any attached files.

**Do you want your identity to be public for this peer review?** For information about this choice, including consent withdrawal, please see our Privacy Policy .

Reviewer #1: No

Reviewer #2: No

Reviewer #3: No

**Figure resubmission:**
---

## [Editor Report · Decision Letter 1]

28 Oct 2025

Dear Dr Riccomagno,

We are pleased to inform you that your manuscript entitled "Functional Role for Cas Cytoplasmic Adaptor Proteins During Cortical Axon Pathfinding" has been editorially accepted for publication in PLOS Genetics. Congratulations!

Yours sincerely,

Ophir Klein

Academic Editor

PLOS Genetics

Fengwei Yu

Section Editor

PLOS Genetics

Aimée Dudley

Editor-in-Chief

PLOS Genetics

Anne Goriely

Editor-in-Chief

PLOS Genetics

BlueSky: @plos.bsky.social

Comments from the reviewers (if applicable):

**Data Deposition**

http://datadryad.org/submit?journalID=pgenetics&manu=PGENETICS-D-25-00757R1

**Press Queries**

---

## [Editor Report · Acceptance letter]

PGENETICS-D-25-00757R1

 Functional Role for Cas Cytoplasmic Adaptor Proteins During Cortical Axon Pathfinding

Dear Dr Riccomagno,

We are pleased to inform you that your manuscript entitled " 

 Functional Role for Cas Cytoplasmic Adaptor Proteins During Cortical Axon Pathfinding" has been formally accepted for publication in PLOS Genetics! Your manuscript is now with our production department and you will be notified of the publication date in due course.

With kind regards,

Zsofia Freund

PLOS Genetics

On behalf of:
